# Privately Publishable Per-instance Privacy

**Rachel Redberg**
Department of Computer Science
UC Santa Barbara
Santa Barbara, CA 93106
rredberg@ucsb.edu

**Yu-Xiang Wang**
Department of Computer Science
UC Santa Barbara
Santa Barbara, CA 93106
yuxiangw@cs.ucsb.edu

## Abstract

We consider how to privately share the personalized privacy losses incurred by objective perturbation, using per-instance differential privacy (pDP). Standard differential privacy (DP) gives us a worst-case bound that might be orders of magnitude larger than the privacy loss to a particular individual relative to a fixed dataset. The pDP framework provides a more fine-grained analysis of the privacy guarantee to a target individual, but the per-instance privacy loss itself might be a function of sensitive data. In this paper, we analyze the per-instance privacy loss of releasing a private empirical risk minimizer learned via objective perturbation, and propose a group of methods to privately and accurately publish the pDP losses at little to no additional privacy cost.

## 1 Introduction

An explosion of data has fueled innovation in machine learning applications and demanded, in equal turn, privacy protection for the sensitive data with which machine learning practitioners train and evaluate models.

Differential privacy (DP) (Dwork et al., 2006, 2014a) has become a mainstay of privacy-preserving data analysis, replacing less robust privacy definitions such as $k$-anonymity which fail to protect against sufficiently powerful de-anonymization attacks (Narayanan & Shmatikov, 2008). In contrast, DP offers provable privacy guarantees that are robust against an arbitrarily strong adversary.

The data curator could trivially protect against privacy loss by reporting a constant function, or by releasing only data-independent noise. The key challenge of DP is to release privatized output that retains utility to the data analyst.

A desired level of utility in a machine learning application might necessitate a high value of $\epsilon$, but the privacy guarantees degrade quickly past $\epsilon = 1$. (Triastcyn & Faltings, 2020) construct an example whereby a differentially private algorithm with $\epsilon = 2$ allows an attacker to use a maximum-likelihood estimate to conclude with up to 88% accuracy that an individual is in a dataset. For $\epsilon = 5$, the theoretical upper bound on the accuracy of an optimal attack is 99.3%.

Moreover, practical applications of differential privacy commonly use large values of $\epsilon$. A study of Apple's deployment of differential privacy revealed that the overall daily privacy loss permitted by the system was as high as $\epsilon = 6$ for Mac OS 10.12.3 and $\epsilon = 14$ for iOS 10.1.1 (Tang et al., 2017) – offering only scant privacy protection!

Recent work (Yu et al., 2021) has empirically justified large privacy parameters by conducting membership inference attacks to demonstrate that these seemingly tenuous privacy guarantees are actually much stronger in practice. These results are unsurprising from the perspective that DP gives a data-independent bound on the worst-case privacy loss which is likely to be a conservative estimate of the risk to a particular individual when a DP algorithm is applied to a particular input dataset.

35th Conference on Neural Information Processing Systems (NeurIPS 2021).

*Per-instance differential privacy* provides a theoretically sound alternative to the empirical approach for revealing the gap between the worst-case DP bound and the actual privacy loss in practice. The privacy loss to a particular individual relative to a fixed dataset might be orders of magnitude smaller than the worst-case bound guaranteed by standard DP. In this case, an algorithm meeting a desired level of utility but providing weak DP guarantees may, for the same level of utility, achieve drastically more favorable *per-instance* DP guarantees.

The remaining challenge is that the per-instance privacy loss is a function of the entire dataset; publishing it directly would negate the purpose of privately training a model in the first place! In this paper, we propose a methodology to privately release the per-instance privacy losses associated with private empirical risk minimization. Our contributions are as follows:

- We introduce *ex-post* per-instance differential privacy to provide a sharp characterization of the privacy loss to a particular individual that adapts to both the input dataset and the algorithm's output.

- We present a novel analysis of the *ex-post* per-instance privacy losses incurred by the objective perturbation mechanism, demonstrating that these *ex-post* pDP losses are orders of magnitude smaller than the worst-case guarantee of differential privacy.

- We propose a group of methods to privately and accurately release the *ex-post* pDP losses. In the particular case of generalized linear models, we show that we can accurately publish the private *ex-post* pDP losses using a dimension- and dataset-independent bound.

- One technical result of independent interest is a new DP mechanism that releases the Hessian matrix by adding a *Gaussian Orthogonal Ensemble* matrix, which improves the classical "AnalyzeGauss" (Dwork et al., 2014b) by roughly a constant factor of 2.

### 1.1 Related Work

This paper builds upon (Wang, 2019), which proposed the per-instance DP framework and left as an open question the matter of publishing the pDP losses. We extend the pDP framework to an *ex-post* setting to provide privacy guarantees that adapt even more fluidly to data-dependent properties of our algorithms. Another fundamental ingredient in our privacy analysis is the objective perturbation algorithm (`Obj-Pert`) of (Chaudhuri et al., 2011), further analyzed by (Kifer et al., 2012), which privately releases the minimizer of an empirical risk by adding a linear perturbation to the objective function before optimizing.

Per-instance DP and *ex-post* per-instance DP belong to a growing family of DP definitions that provide a more fine-grained characterization of the privacy loss. Among these are data-dependent DP (Papernot et al., 2018), which conditions on a fixed dataset; personalized DP (Ghosh & Roth, 2011; Ebadi et al., 2015; Liu et al., 2015), which conditions on a fixed individual's datapoint; and *ex-post* DP (Ligett et al., 2017), which conditions on the realized output of the algorithm. Per-instance DP conditions on both a fixed dataset and a fixed individual's datapoint, and *ex-post* per-instance DP adapts even further to the realized output of the algorithm. A more detailed comparison of these DP variants is included in the supplementary materials.

Other data-adaptive methodologies include propose-test-release (Dwork & Lei, 2009) and local sensitivity (Nissim et al., 2007). In addition, Bayesian differential privacy (Triastcyn & Faltings, 2020) provides data-dependent privacy guarantees that afford strong protection to "typical" data by making distributional assumptions about the sensitive data. The Rényi-DP-based privacy filters of (Feldman & Zrnic, 2020) are also closely related to our work; the authors study composition of personalized (but not per-instance) privacy losses using adaptively-chosen privacy parameters.

## 2 Preliminaries

### 2.1 Symbols and Notation

We write the output of a randomized algorithm $\mathcal{A}$ as $\mathcal{A}(\cdot)$, and for continuous distributions we take $\Pr[\mathcal{A}(D) = o]$ to be the value of the probability density function at output $o$.

We will let $z \in \mathcal{Z}$ refer to both an individual and their data; for example, individual $z$ holds data $z = (x, y)$ in a supervised learning problem. We take $\mathcal{Z}^* = \cup_{n=0}^{\infty} \mathcal{Z}^n$ to be the space of datasets with

an unspecified number of data points. $D_{\pm z} \in \mathcal{Z}^*$ denotes the fixed dataset $D = \{z_1, \ldots, z_n\} \in \mathcal{Z}^*$ with the data point $z$ removed from $D$ if $z \in D$, or added to $D$ if $z \notin D$. In our mathematical expressions, we use "$\pm$" to mean "add if $z \notin D$, subtract otherwise". Similarly. "$\mp$" means "subtract if $z \notin D$, add otherwise".

We distinguish between $\epsilon$ as fixed input to a DP algorithm, and $\epsilon(\cdot)$ as a function parameterized according to a particular DP relaxation — e.g., $\epsilon(o, D, D_{\pm z})$ means the *ex-post* per-instance privacy loss conditioned on output $o$, dataset $D$, and data point $z$.

## 2.2 Differential Privacy

Let $\mathcal{Z}$ denote the data domain, and $\mathcal{R}$ the set of all possible outcomes of algorithm $\mathcal{A}$. Fix $\epsilon, \delta \geq 0$.

**Definition 1.** (Differential privacy) A randomized algorithm $\mathcal{A} : \mathcal{Z}^* \to \mathcal{R}$ satisfies $(\epsilon, \delta)$-DP if for all datasets $D \in \mathcal{Z}^*$ and data points $z \in \mathcal{Z}$, and for all measurable sets $S \subset \mathcal{R}$,

$$\Pr\big[\mathcal{A}(D) \in S\big] \leq e^\epsilon \Pr\big[\mathcal{A}(D_{\pm z}) \in S\big] + \delta.$$

Differential privacy guarantees that the presence or absence of any particular data record has little impact on the output distribution of a randomized algorithm. In this paper we use the "add/remove" notion of DP, by which we construct neighboring dataset $D_{\pm z}$ by adding or removing an individual $z$ from dataset $D$.

DP is powerful and universal in that its guarantee applies to any $D, z$ and set of output events. However, there are often situations where the privacy losses of $\mathcal{A}$ vary drastically depending on its input data, and the privacy loss bound $\epsilon$ (protecting even the worst-case pair of neighboring datasets) may not be informative of the privacy loss incurred to individuals when the input to $\mathcal{A}$ is typical. This motivated (Wang, 2019) to consider a per-instance version of the DP definition.

**Definition 2.** (Per-instance differential privacy) A randomized algorithm $\mathcal{A} : \mathcal{Z}^* \to \mathcal{R}$ satisfies $\big(\epsilon(D, D_{\pm z}), \delta\big)$-pDP if for dataset $D$ and data point $z$, and for all measurable sets $S \subset \mathcal{R}$,

$$\Pr\big[\mathcal{A}(D) \in S\big] \leq e^\epsilon \Pr\big[\mathcal{A}(D_{\pm z}) \in S\big] + \delta,$$
$$\Pr\big[\mathcal{A}(D_{\pm z}) \in S\big] \leq e^\epsilon \Pr\big[\mathcal{A}(D) \in S\big] + \delta.$$

The pDP definition can be viewed as using a function $\epsilon(D, D_{\pm z})$ that more precisely describes the privacy guarantee in protecting a fixed data point $z$ when $\mathcal{A}$ is applied to dataset $D$.

As it turns out, it is most convenient for us to work with an even more *instance-specific* description of the privacy loss that is further parameterized by the realized output of $\mathcal{A}$ *ex-post* — after the random coins of $\mathcal{A}$ are flipped and the outcome released.

**Definition 3.** (*Ex-post* per-instance differential privacy) A randomized algorithm $\mathcal{A}$ satisfies $\epsilon(\cdot)$-*ex-post* per-instance differential privacy for an individual $z$ and a fixed dataset $D$ at an outcome $o \in \mathrm{Range}(\mathcal{A})$ if

$$\left| \log \left( \frac{\Pr\big[\mathcal{A}(D) = o\big]}{\Pr\big[\mathcal{A}(D_{\pm z}) = o\big]} \right) \right| \leq \epsilon(o, D, D_{\pm z}).$$

This definition generalizes the *ex-post* DP definition (Ligett et al., 2017) (introduced for a different purpose) to a *per-instance* version that depends on a given pair of neighboring datasets. The above quantity is essentially the absolute value of the log-odds ratio, used extensively in hypothesis testing. Intuitively, the *ex-post* per-instance privacy loss $\epsilon(o, D, D_{\pm z})$ describes how confidently an attacker could infer, given the output of algorithm $\mathcal{A}$, whether or not individual $z$ is in dataset $D$.

Despite (or perhaps because of) its precise accounting for privacy, *ex-post* pDP could reveal sensitive information about the dataset, as the following example explicitly illustrates.

**Example 4** (The privacy risk of exposing *ex-post* pDP)**.** *Consider a standard Gaussian mechanism $\mathcal{A}$ that adds noise to a counting query $Q$ applied to dataset $D$, i.e. $\mathcal{A}(D) = Q(D) + \mathcal{N}(0, \sigma^2)$. $Q$ has global sensitivity $\Delta_Q = 1$. We will show that an attacker, knowing only the output $o$ of algorithm $\mathcal{A}$, her* ex-post *pDP loss and that her individual data is not contained in dataset $D$, can conclusively uncover the sensitive quantity $Q(D)$ protected by algorithm $\mathcal{A}$.*

*Following the proof of Theorem* **??**, *the* ex-post *pDP can be directly calculated as*

$$\epsilon(o, D, D_{\pm z}) = \frac{|Q(D) - Q(D_{\pm z})| |2o - Q(D) - Q(D_{\pm z})|}{2\sigma^2}.$$

*Enter attacker $z$, who has auxiliary information: she knows that her own individual data is not contained in $D$. After algorithm $\mathcal{A}$ is applied to $D$, attacker $z$ receives output $o = 1$ and is informed of her* ex-post *pDP $\epsilon(o, D, D_{+z})$. Since $Q(D_{+z}) = Q(D) + 1$ is known, attacker $z$ can solve for $Q(D)$ and obtain $Q(D) = o - 0.5 \pm \sigma^2 \epsilon(o, D, D_{+z})$. With probability 1, only one of the two possibilities is an integer[1]. Therefore, exposing* ex-post *pDP in this case completely reveals $Q(D)$.*

**Problem statement.** The lesson of Example 4 is that we cannot directly reveal the *ex-post* pDP losses without potentially nullifying the algorithm's privacy benefits. How, then, can we privately and accurately publish the *ex-post* pDP losses?

The goal of this paper is to develop an algorithm that publishes a *function* $\tilde{\epsilon} : \mathcal{Z} \to \mathbb{R}$ whose output estimates the *ex-post* pDP loss to an individual $z$ of releasing the output $\hat{\theta}^P$ from the objective perturbation mechanism. Any individual (not just those whose data is contained in the dataset) can plug her own data $z$ into this function in order to receive a high-probability bound on her *ex-post* pDP loss which does not depend directly on any sensitive data except her own.

This requirement offers the same type of privacy protection as joint differential privacy (Kearns et al., 2014), which relaxes the standard DP definition by allowing an algorithm's output to individual $z$ to be sensitive only in her own private data. Our notion of privacy is slightly more general in that it holds for individuals both in and out of the dataset. The difference lies in how the algorithm's output space is defined; whereas a joint DP algorithm produces a fixed-length tuple partitioning the output to each individual in the dataset, our algorithm outputs a function whose domain includes any data point $z \in \mathcal{Z}$. As a result, our methods are robust against collusion by arbitrary coalitions of adversaries, allowing repeated queries by any group of individuals without invalidating the privacy guarantees promised by the pDP losses.

## 2.3 Problem Setting

We consider a general family of problems known as *private empirical risk minimization* (ERM), which aim to approximate the solution to an ERM problem while preserving privacy. That is, we wish to privately solve optimization problems of the form

$$\hat{\theta} = \operatorname*{argmin}_{\theta \in \Theta} L(\theta; D) + r(\theta),$$

where $r(\theta)$ is a regularizer and $L(\theta; D) = \sum_{i=1}^{n} \ell(\theta; z_i)$ a loss function. Throughout, we assume that $\ell(\theta; z)$ and $r(\theta)$ are convex and twice-differentiable with respect to $\theta$. Dataset $D$ is given by $D = \{z_i\}_{i=1}^{n}$, and $z_i = (x_i, y_i)$ for $x_i \in \mathcal{X} \subseteq \mathbb{R}^d$ and $y \in \mathcal{Y} \subseteq \mathbb{R}$, where $||x||_2 \leq 1$ and $|y| \leq 1$. We consider only unconstrained optimization over $\Theta = \mathbb{R}^d$.

## 2.4 Objective Perturbation

The objective perturbation algorithm solves

$$\hat{\theta}^P = \operatorname*{argmin}_{\theta \in \Theta} L(\theta; D) + r(\theta) + \frac{\lambda}{2} ||\theta||_2^2 + b^T \theta, \tag{1}$$

where $b \sim \mathcal{N}(0, \sigma^2 I_d)$ and parameters $\sigma, \lambda$ are chosen according to a desired $(\epsilon, \delta)$-DP guarantee.

---

**Algorithm 1** Release $\hat{\theta}^P$ via `Obj-Pert` (Kifer et al., 2012)

---

**Input:** Dataset $D$, noise parameter $\sigma$, regularization parameter $\lambda$, loss function $L(\theta; D) = \sum_i \ell(\theta; z_i)$, convex and twice-differentiable regularizer $r(\theta)$, convex set $\Theta$.
**Output:** $\hat{\theta}^P$, the minimizer of the perturbed objective.
Draw noise vector $b \sim \mathcal{N}(0, \sigma^2 I)$.
Compute $\hat{\theta}^P$ according to (1).

---

[1]Take $Q(D) = 0$ and $o = 0.1$ as an example, the two possibilities are 0 and $-0.8$.

**Theorem 5** (Privacy guarantees of Algorithm 1 (Kifer et al., 2012)). *Consider dataset $D = \{z_i\}_{i=1}^n$; loss function $L(\theta; D) = \sum_i \ell(\theta; z_i)$; convex regularizer $r(\theta)$; and convex domain $\Theta$. Assume that $\nabla^2 \ell(\theta; z_i) \prec \beta I_d$ and $||\nabla \ell(\theta; z_i)||_2 \leq \xi$ for all $z_i \in \mathcal{X} \times \mathcal{Y}$ and for all $\theta \in \Theta$. For $\lambda \geq 2\beta/\epsilon_1$ and $\sigma = \xi^2(8\log(2/\delta) + 4\epsilon_1)/\epsilon_1^2$, Algorithm 1 satisfies $(\epsilon_1, \delta)$-differential privacy.*

The privacy guarantees stated in Theorem 5 apply even when $\theta$ is constrained to a closed convex set, but for ease of our per-instance privacy analysis we will require $\Theta = \mathbb{R}^d$ from this point on.

## 3 Privately Publishable pDP

### 3.1 pDP Analysis of Objective Perturbation

Our goal in this section is to derive the personalized privacy losses (under Definition 3) associated with observing the output $\hat{\theta}^P$ of objective perturbation. This *ex-post* perspective is highly adaptive and also convenient for our analysis of Algorithm 1, whose privacy parameters are a function of the data. Since we are analyzing the per-instance privacy cost of *releasing* $\hat{\theta}^P$, it makes perfect sense to condition the pDP loss on the privatized output of the computation.

Our first technical result is a precise calculation of the *ex-post* pDP loss of objective perturbation.

**Theorem 6** (*ex-post* pDP loss of objective perturbation for a convex loss function). *Let $J(\theta; D) = L(\theta; D) + r(\theta) + \frac{\lambda}{2}||\theta||_2^2$ such that $L(\theta; D) + r(\theta) = \sum_i \ell(\theta; z_i) + r(\theta)$ is a convex and twice-differentiable regularized loss function, and sample $b \sim \mathcal{N}(0, \sigma^2 I_d)$. Then for every privacy target $z = (x, y)$, releasing $\hat{\theta}^P = \mathrm{argmin}_{\theta \in \mathbb{R}^d} J(\theta; D) + b^T \theta$ satisfies $\epsilon_1(\hat{\theta}^P, D, D_{\pm z})$-ex-post per-instance differential privacy with*

$$\epsilon_1(\hat{\theta}^P, D, D_{\pm z}) = \left| -\log \prod_{j=1}^d \left(1 \mp \mu_j\right) + \frac{1}{2\sigma^2}||\nabla \ell(\hat{\theta}^P; z)||_2^2 \pm \frac{1}{\sigma^2}\nabla J(\hat{\theta}^P; D)^T \nabla \ell(\hat{\theta}^P; z)\right|,$$

*where $\mu_j = \lambda_j u_j^T \left(\nabla b(\hat{\theta}^P; D) \mp \sum_{k=1}^{j-1} \lambda_k u_k u_k^T\right)^{-1} u_j$ according to the eigendecomposition $\nabla^2 \ell(\theta; z) = \sum_{k=1}^d \lambda_k u_k u_k^T$.*

*Proof sketch.* Following the analysis of (Chaudhuri et al., 2011), we establish a bijection between the mechanism output $\hat{\theta}^P$ and the noise vector $b$, and use a change-of-variables defined by the Jacobian mapping between $\hat{\theta}^P$ and $b$ in order to rewrite the log-probability ratio in terms of the probability density function of $b$. First-order conditions then allow us to solve directly for the distribution of $b$. To calculate the first term of the above equation, we use the eigendecomposition of the Hessian $\nabla^2 \ell(\hat{\theta}^P; z)$ and recursively apply the matrix determinant lemma. The rest of the proof is straightforward algebra. The full proof is given in Appendix **??**. □

The above expression holds for any convex loss function, but is a bit unwieldy. The calculation becomes much simpler when we assume $\ell(\cdot)$ to be a generalized linear loss function, with inner-product form $\ell(\theta; z) = f(x^T\theta; y)$. For the sake of interpretability, we will defer further discussion of the *ex-post* pDP loss of objective perturbation until after presenting the following corollary.

**Corollary 7** (*ex-post* pDP loss of objective perturbation for GLMs). *Let $J(\theta; D) = L(\theta; D) + r(\theta) + \frac{\lambda}{2}||\theta||_2^2$ such that $L(\theta; D) = \sum_i \ell(\theta; z_i)$ is a linear loss function, and sample $b \sim \mathcal{N}(0, \sigma^2 I_d)$. Then for every privacy target $z = (x, y)$, releasing $\hat{\theta}^P = \mathrm{argmin}_{\theta \in \mathbb{R}^d} J(\theta; D) + b^T \theta$ satisfies $\epsilon_1(\hat{\theta}^P, D, D_{\pm z})$-ex-post per-instance differential privacy with*

$$\epsilon(\hat{\theta}^P, D, D_{\pm z}) \leq \left| -\log\left(1 \pm f''(\cdot)\mu(x)\right) + \frac{1}{2\sigma^2}||\nabla \ell(\hat{\theta}^P; z)||_2^2 \pm \frac{1}{\sigma^2}\nabla J(\hat{\theta}^P; D)^T \nabla \ell(\hat{\theta}^P; z)\right|,$$

*where $\mu(x) = x^T\left(\nabla^2 J(\hat{\theta}^P; D)\right)^{-1}x$, $\nabla \ell(\hat{\theta}^P; z) = f'(x^T\hat{\theta}^P; y)x$ and $f''(\cdot)$ is shorthand for $f''(\cdot) = f''(x^T\hat{\theta}^P; y)$. The notation $b(\hat{\theta}^P; D)$ means the realization of the noise vector $b$ for which the output of Algorithm 1 will be $\hat{\theta}^P$ when the input dataset is $D$.*

Note that the quantity $\mu(x)$ in the first term is the *generalized leverage score* (Wei et al., 1998), quantifying the influence of a data point on the model fit. The second and third terms are a function of the gradient of the loss function and provide a complementary measure of how well the fitted model predicts individual $z$'s data.

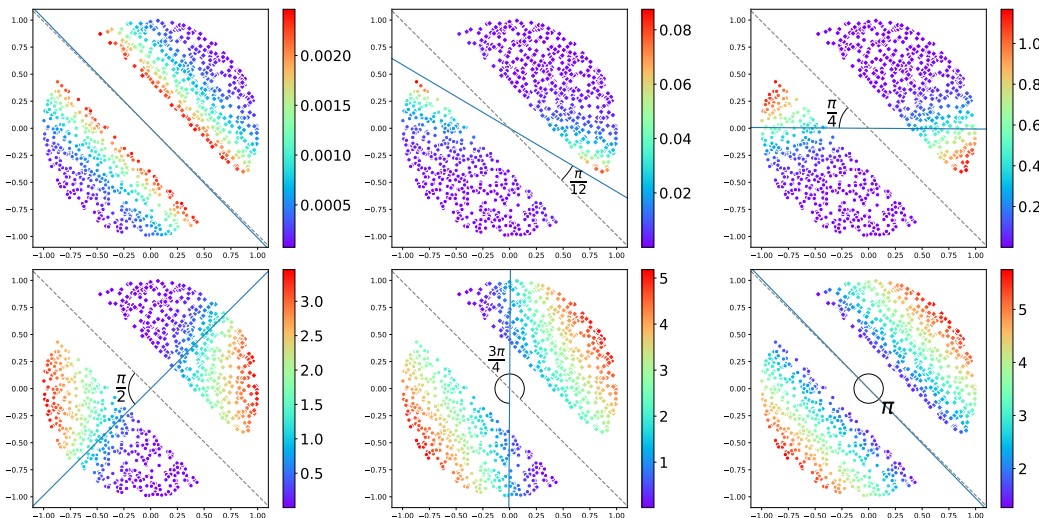

Figure 1: Visualization of *ex-post* pDP losses for logistic regression ($n = 1000, d = 2$).

Since the *ex-post* pDP is a function of $\hat{\theta}^P$, we don't even need to run Algorithm 1 to calculate *ex-post* pDP losses – we can plug in directly to Corollary 7 in order to calculate the pDP distribution induced by any hypothetical $\hat{\theta}^P$. For Figure 1, we use a synthetic dataset $D$ sampled from the unit ball with two linearly separable classes separated by margin $m = 0.4$. Then we solve for $\hat{\theta} = \operatorname{argmin} J(\theta; D)$ with $\lambda = 1$ to minimize the logistic loss, and directly perturb the output by rotating it by angle $\omega \in [0, \frac{\pi}{12}, \frac{\pi}{4}, \frac{\pi}{2}, \frac{3\pi}{4}, \pi]$. We then denote $\hat{\theta}^P := \theta_{+\omega}$ to mean $\theta$ rotated counter-clockwise by angle $\omega$. The color scale is a function of the *ex-post* pDP loss of data point $z$.

Figure 1 illustrates how the mechanism output $\hat{\theta}^P$ affects the *ex-post* pDP distribution of objective perturbation for our logistic regression problem. For $\omega \in [0, \frac{\pi}{12}]$, the data points closest to the decision boundary have the highest *ex-post* pDP loss. These data points have a strong effect on the learned model and would therefore have high *leverage scores*, making the first term dominate. As the perturbation (and model error) increases, the second and third terms dominate; the more badly a model predicts a data point, the less protection this data point has.

Hidden in this analysis are the $\delta$'s of Theorem 5, which along with the choice of $\sigma$ and $\lambda$ could affect which of the three terms is dominant. Fortunately, the probability of outputting something like $\hat{\theta}^P = \theta_{+\pi}$ is astronomically low for any reasonable privacy setting!

## 3.2 Releasing the pDP losses

Next we consider: after having released $\hat{\theta}^P$ and calculated the per-instance privacy losses of doing so, how do we privately release these pDP losses? Our goal is to allow any individual $z \in \mathcal{Z}$ (in the dataset or not) to know her privacy loss while preserving the privacy of others in the dataset.

Observe that the expression from Theorem 6 depends on the dataset $D$ only through two quantities: the leverage score $\mu(x) = x^T \left(\nabla^2 J(\hat{\theta}^P; D)\right)^{-1} x$ and the inner product $\nabla J(\hat{\theta}^P; D)^T \nabla \ell(\hat{\theta}^P; z)$. As a result, if we can find a data-independent bound for these two terms, or privately release them with only a small additional privacy cost, then we are done.

### 3.2.1 Data-independent bound of *ex-post* pDP losses

Below, we present a pair of lemmas which will allow us to find a high-probability, data-independent bound on the *ex-post* pDP loss.

**Theorem 8.** *Suppose $\ell(\cdot)$ is a function with continuous second-order partial derivatives. Then*

$$\left| -\log \prod_{j=1}^{d} \left( 1 \mp \mu_j \right) \right| \le -\sum_{j=1}^{d} \log(1 - \frac{\lambda_j}{\lambda}),$$

*where $\mu_j = \lambda_j u_j^T \left( \nabla \mathbf{b}(\hat{\theta}^P; D) \mp \sum_{k=1}^{j-1} \lambda_k u_k u_k^T \right)^{-1} u_j$ according to the eigendecomposition $\nabla^2 \ell(\hat{\theta}^P; z) = \sum_{k=1}^{d} \lambda_k u_k u_k^T$. When specializing to linear loss functions such that $\ell(\theta; z) = f(x^T \theta; y)$, $\lambda_j = 0$ for all $j > 1$ and the above bound can be simplified to $-\log \left( 1 - f''(x^T \hat{\theta}^P; y) ||x||_2^2 / \lambda \right)$.*

**Theorem 9.** *Let $\hat{\theta}^P$ be a random variable such that $\hat{\theta}^P = \arg\min \left( J(\theta; D) + b^T \theta \right)$ as in (1), where $b \sim \mathcal{N}(0, \sigma^2 I_d)$ and $\ell(\theta; z)$ is a convex and twice-differentiable loss function. Then for $z \in \mathcal{Z}$, the following holds with probability $1 - \rho$:*

$$\left| \nabla J(\hat{\theta}^P; D)^T \nabla \ell(\hat{\theta}^P; z) \right| \le \sigma \sqrt{2 \log(2d/\rho)} \|\nabla \ell(\hat{\theta}^P; z)\|_1.$$

*For linear loss functions the bound can be substantially strengthened to*

$$\left| \nabla J(\hat{\theta}^P; D)^T \nabla \ell(\hat{\theta}^P; z) \right| \le f'(x^T \hat{\theta}^p; y) \sigma ||x||_2 \sqrt{2 \log(2/\rho)}.$$

We make a few observations on the bounds. First, the general bound in Theorem 9 holds simultaneously for all $z$ and it depends only logarithmically in dimension when the features are *sparse*. Second, the bound for a linear loss function is dimension-free and somewhat surprising because we are actually bounding an inner product of two *dependent* random vectors (both depend on $\hat{\theta}^P$).

Finally, we remark that the bounds in this section are data-independent in that they do not depend on the rest of the dataset beyond already released information $\hat{\theta}^P$. It allows us to reveal a pDP bound of each individual when she plugs in her own data without costing any additional privacy budget!

### 3.3 The privacy report

For certain regimes, we may wish to consider privatizing the data-dependent quantities of the *ex-post* pDP losses, at an additional privacy cost, as an alternative to using data-independent bounds. Of course, it only makes sense to do so if we can show that (a) these data-dependent estimates are more accurate than the data-independent bounds; (b) the overhead of releasing additional quantities (the additional privacy cost in terms of both DP and pDP) is not too large; and (c) we can share the pDP losses of the private reporting algorithm using data-independent bounds (so we do not have to recursively publish such reports).

Full details are in the appendix. We show that by adding slightly more regularization than required by `Obj-Pert` (i.e., making $\lambda$ just a bit larger so that the minimum eigenvalue of the Hessian $H = \nabla^2 J$ is above a certain threshold), we can find a multiplicative bound that estimates $\mu(x) = x^T H^{-1} x$ uniformly for all $x$. We do so by adding noise to the Hessian using a natural variant of "Analyze Gauss" (Dwork et al., 2014b), hence privately releasing $\overline{\mu^P} : \mathcal{X} \to \mathbb{R}$. See Algorithm 2 for details.

For brevity, we use the short-hands $f'(\cdot) := f'(x^T \hat{\theta}^P; y)$ and $f''(\cdot) := f''(x^T \hat{\theta}^P; y)$, where $\ell(\theta; z) = f(x^T \theta; y)$ for GLMs. $F_{\mathcal{N}(0,1)}^{-1}$ is the inverse CDF of the standard normal distribution, and $F_{GOE(d)}^{-1}$ is the inverse CDF of the largest eigenvalue of the Gausian Orthogonal Ensemble (GOE) matrix, whose distribution is calculated exactly by (Chiani, 2014). Algorithm 2 specializes to GLMs for clarity of presentation, but we could adapt it to any convex loss function by replacing the GLM-specific bounds with the more general ones.

We implicitly assume that the data analyst has already decided the privacy budgets $\epsilon_2$ and $\epsilon_3$ for the data-dependent release of the gradient (third term of $\epsilon_1(\cdot)$) and of the Hessian (first term of $\epsilon_1(\cdot)$). Inputs $\sigma_2$ and $\sigma_3$ are then calibrated to achieve $(\epsilon_2, \rho)$-DP and $(\epsilon_3, \rho)$-DP, respectively.

---

**Algorithm 2** Privacy report for `Obj-Pert` on GLMs

---

**Input:** $\hat{\theta}^p \in \mathbb{R}^d$ from `Obj-Pert`, noise parameter $\sigma, \sigma_2, \sigma_3$; regularization parameter $\lambda$; Hessian $H := \sum_i \nabla^2 \ell(\hat{\theta}^p; z_i) + \lambda I_d$, Boolean B $\in$ [DATA-INDEP, DATA-DEP], failure probability $\rho$

**Require:** $\lambda \geq 2\sigma_3 F^{-1}_{\lambda_1(\text{GOE}(d))}(1 - \rho/2)$

**Output:** Reporting function $\tilde{\epsilon} : (x, y), \delta \to \mathbb{R}^3_+$

**if** B = DATA-INDEP **then**

    Set $\epsilon_2(\cdot) := 0, \epsilon_3(\cdot) := 0$.

    Set $\overline{g^P}(z) := \sigma\|f'(\cdot)x\|_2 F^{-1}_{\mathcal{N}(0,1)}(1 - \rho/2)$ and set $\overline{\mu^p}(x) := \frac{\|x\|^2}{\lambda}$.

**else if** B = DATA-DEP **then**

    Privately release $\hat{g}^p$ by Algorithm **??** with parameter $\sigma_2$.

    Set $\epsilon_2(\cdot)$ according to Theorem **??**.

    Set $\overline{g^P}(z) := \min\left\{ f'(\cdot)[\hat{g}^P(z)]^T x + \sigma_2\|f'(\cdot)x\|_2 F^{-1}_{\mathcal{N}(0,1)}(1 - \rho/2), \ \sigma\|f'(\cdot)x\|_2 F^{-1}_{\mathcal{N}(0,1)}(1 - \rho/2) \right\}$.

    Privately release $\hat{H}^p$ by a variant of "Analyze Gauss"[2] with parameter $\sigma_3$.

    Set $\epsilon_3(\cdot)$ according to Statement 2 of Theorem 10.

    Set $\overline{\mu^p}(x) = \frac{3}{2} x^T [\hat{H}^p]^{-1} x$.

**end if**

Set $\overline{\epsilon_1^p}(z) := \left| -\log\left(1 - f''(\cdot)\overline{\mu^p}(x)\right) \right| + \frac{\|f'(\cdot)x\|_2^2}{2\sigma^2} + \frac{\left|\overline{g^P}(z)\right|}{\sigma^2}$.

Output the function $\tilde{\epsilon}(z) := \left(\overline{\epsilon_1^p}(z), \epsilon_2(z), \epsilon_3(z)\right)$.

---

Note that the pDP functions $\epsilon_2(\cdot)$ and $\epsilon_3(\cdot)$ – which we use to report the additional pDP losses of releasing the private estimates of the gradient and the Hessian – do not depend on the dataset, and thus are not required to be separately released. The privately released pDP functions depend on $\hat{\theta}^P$; to reduce clutter, we omit this parameter in our presentation of Algorithm 2.

**Theorem 10.** *There is a universal constant $C$ such that if $\lambda > C\sigma_2\sqrt{d}(1 + (\log(1/\rho))^{2/3})$, then Algorithm 2 satisfies the following properties*

1. $\left(\frac{\xi^2}{2\sigma_2^2} + \frac{\beta^2}{4\sigma_3^2} + \sqrt{\frac{\xi^2}{\sigma_2^2} + \frac{\beta^2}{2\sigma_3^3}}\sqrt{2\log(1/\delta)}, \delta\right)$-*DP*

2. $\left(\frac{f'(\hat{\theta}^p;z)^2\|x\|^2}{2\sigma_2^2} + \frac{f''(\hat{\theta}^p;z)^2\|x\|^4}{4\sigma_3^2} + \sqrt{\frac{f'(\hat{\theta}^p;z)^2\|x\|^2}{\sigma_2^2} + \frac{f''(\hat{\theta}^p;z)^2\|x\|^4}{2\sigma_3^2}}\sqrt{2\log(1/\delta)}, \delta\right)$-*pDP for all $x \in \mathcal{X}$ and $0 \leq \delta < 1$.*

3. *For a fixed input $z$ and $D$, and all $\rho > 0$, the privately released privacy report $\tilde{\epsilon}(\cdot)$ satisfies that $\epsilon_1(\hat{\theta}^p, D, D_{\pm z}) \leq \overline{\epsilon_1^p}(z) \leq 12\epsilon_1(\hat{\theta}^p, D, D_{\pm z}) + \frac{|f'(\cdot)|\|x\|}{\sigma_2}\sqrt{2\log(2/\rho)}$ with probability $1 - 3\rho$ where $\epsilon_1(\cdot)$ is the expression from Theorem 6.*

**Accurate approximation with low privacy cost.** This theorem shows that if we use a slightly larger $\lambda$ in ObjPert then we get an upper bound of the pDP for each individual $z$ up to a multiplicative and an additive factor. The multiplicative factor is coming from a multiplicative approximation of $-\log\left(1 \pm f''(\cdot)\mu(x)\right)$ and the additive error is due to the additional noise added for releasing the third term $\frac{1}{\sigma^2}\nabla J(\hat{\theta}^P; D)^T \nabla\ell(\hat{\theta}^P; z)$. The additional DP and pDP losses for releasing $H$ and $g$ are comparable to the DP and pDP losses in Objective Perturbation itself if $\sigma_2 \asymp \sigma_3 \asymp \sigma$.

Moreover, while using a large $\lambda$ may appear to introduce additional bias, the required choice of $\lambda \asymp \sqrt{d}\sigma$ is actually exactly the choice to obtain the minimax rate in general convex private ERM (Bassily et al., 2014) (Figure 2 demonstrates the impact of increasing $\lambda$).

**Joint DP interpretation.** Finally, we can also interpret our results from a joint-DP perspective (Kearns et al., 2014). Given any realized output $\hat{\theta}^p \in \mathbb{R}^d$, the tuple of $\{\tilde{\epsilon}(z_1, \hat{\theta}^p), ..., \tilde{\epsilon}(z_1, \hat{\theta}^p)\}$

---

[2]Instead of adding "analyze-gauss" noise, we sample from the Gaussian Orthogonal Ensemble (GOE) distribution to obtain a random matrix (Appendix **??**). Under this model we show that $\tau$ is on the order of $O(\sqrt{d}(1 + \log(C/\rho)^{3/2}))$.

satisfies joint DP with the same $\epsilon$ parameter as in Theorem 10. This follows from the billboard lemma (Hsu et al., 2016).

## 4 Experiments

Here we evaluate our methods to release the pDP losses using logistic regression as a case study. In Section 4.1, we demonstrate that the stronger regularization required by Algorithm 2 does not affect the utility of the model. In Section 4.2 we show that by carefully allocating the privacy budget of the data-dependent release, we can achieve a more accurate estimate of the *ex-post* pDP losses of Algorithm 1 compared to the data-independent release, with reasonable overhead (same overall DP budget and only a slight uptick in the overall pDP losses).

Experiments with linear regression, with additional datasets and with alternative privacy budget allocation schemes are included in the supplementary materials.

### 4.1 Stronger regularization does not worsen model utility

In this experiment we use a synthetic dataset generated by sampling $x_i, \theta \sim \mathcal{N}(0, I_d)$ and normalizing each $x_i \in X$ so that $||x_i||_2 = 1$. Then we rescale $Y = X\theta$ to ensure $y_i \in [0, 1]$ for each $y_i \in Y$.

Algorithm 2 requires a larger $\lambda$ than suggested by Theorem 5 in order to achieve a uniform multiplicative approximation of $\mu(\cdot)$. We investigate the effect of stronger regularization on the utility of a private logistic regression model applied to a synthetic dataset ($n = 1000, d = 50$), for several settings of $\epsilon_1$. Since each value of $\epsilon_1$ demands a different minimum value of $\lambda$ in order to achieve $(\epsilon_1, \delta)$-differential privacy, we compare via "$\lambda$-inflation": a measure of how many times larger we set $\lambda$ than its minimum value required to achieve the worst-case DP bound of objective perturbation.

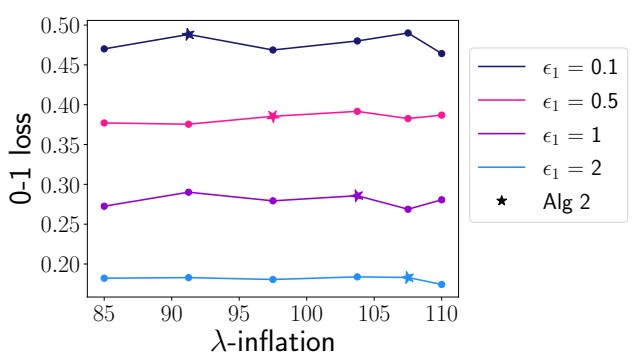

Figure 2: Utility of Obj-Pert with larger $\lambda$.

E.g., for logistic regression the objective perturbation mechanism requires $\lambda \geq \frac{1}{2\epsilon}$, and so in Figure 2 a $\lambda$-inflation value of 10 means that we set $\lambda = \frac{5}{\epsilon}$. For each $\lambda$-inflate value $c$, we run Algorithm 1 with $\lambda = c\lambda_{\texttt{Obj-Pert}}$. In particular, the star symbol marks the level of $\lambda$-inflation enforced by Algorithm 2. The experimental results summarized in Figure 2 show that the performance of the private logistic regression model (as measured by the 0-1 loss) remains roughly constant across varying scales of $\lambda$.

### 4.2 Comparison of data-independent and data-dependent bounds

The following experiments feature the credit card default dataset ($n = 30000, d = 21$) (Yeh & Lien, 2009) from the UCI Machine Learning Repository. We privately train a binary classifier to predict whether or not a credit card client $z$ defaults on her payment (Algorithm 1), and calculate the true pDP loss $\epsilon_1(\cdot)$ as well as the data-independent and -dependent estimates $\overline{\epsilon_1^P}(\cdot)$ for each $z$ in the training set (Algorithm 2).

The failure probabilities for both Algorithms 1 and 2 are set as $\delta = \rho = 10^{-6}$. Our choices of $\sigma$ and $\lambda$ depend on $\epsilon_1$ and follows the requirements stated in Theorem 5 to achieve DP. We don't use any additional regularization, i.e. $r(\theta) = 0$. For the data-dependent release, the noise parameters $\sigma_2, \sigma_3$ are each calibrated according to the analytic Gaussian mechanism of (Balle & Wang, 2018).

Using $\epsilon = 1$ as a DP budget, we investigate how to allocate the privacy budget among the components of the data-dependent release ($\epsilon = \epsilon_1 + \epsilon_2 + \epsilon_3$) to achieve a favorable comparison with the data-independent release which requires no additional privacy cost ($\epsilon = \epsilon_1$). The configuration described in Figure 3, which skews the data-dependent privacy budget toward more accurately releasing $\overline{\epsilon_1^P}(\cdot)$, was empirically chosen as an example where the sum $\overline{\epsilon_1^P}(\cdot) + \epsilon_2(\cdot) + \epsilon_3(\cdot)$ of privately released pDP

losses of the data-dependent approach are comparable to the privately released *ex-post* pDP loss $\overline{\epsilon_1^P}(\cdot)$ of the data-independent approach. Note that $\epsilon_2(\cdot)$ and $\epsilon_3(\cdot)$ aren't *ex-post* in the traditional sense; however, we feel comfortable summing $\overline{\epsilon_1^P}(\cdot) + \epsilon_2(\cdot) + \epsilon_3(\cdot)$ since all three terms are a function of $\hat{\theta}^P$ and individual $z$'s data. Note also that since the total budget $\epsilon$ is the same for both the data-independent and -dependent releases, $\epsilon_1$ differs between them. Therefore Figure **??** compares the accuracy of both approaches using the ratio between $\overline{\epsilon_1^P}(\cdot)$ and $\epsilon_1(\cdot)$ rather than their raw values.

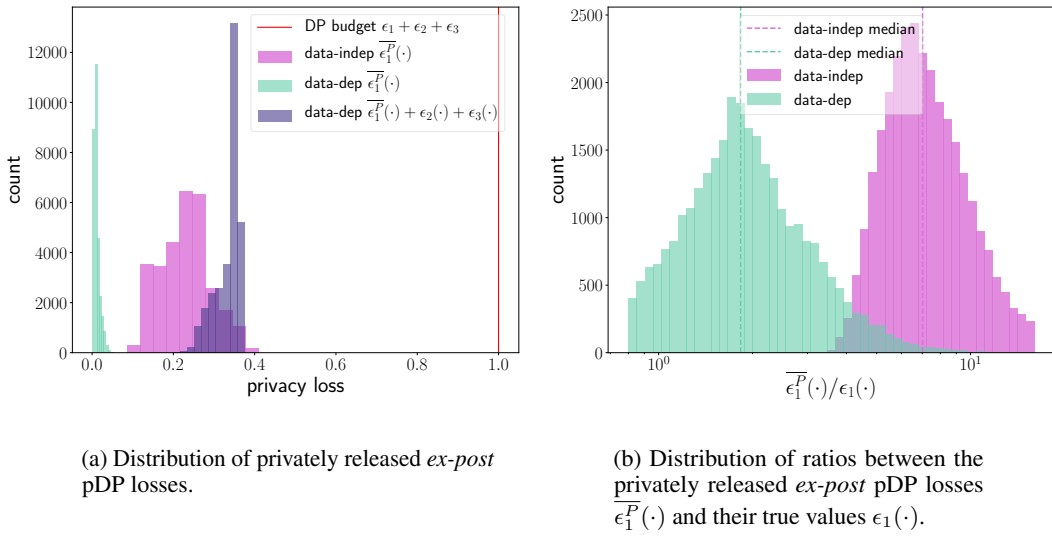

(a) Distribution of privately released *ex-post* pDP losses.

(b) Distribution of ratios between the privately released *ex-post* pDP losses $\overline{\epsilon_1^P}(\cdot)$ and their true values $\epsilon_1(\cdot)$.

Figure 3: True and privately released pDP losses when the total privacy budget is $\epsilon = 1$. For the data-independent release we use the entire privacy budget on releasing $\hat{\theta}^P$ ($\epsilon_1 = 1$). For the data-dependent release we reserve some of the privacy budget for releasing $\overline{\mu^P}(\cdot)$ and $\overline{g^P}(\cdot)$ ($\epsilon_1 = 0.2, \epsilon_2 = 0.7, \epsilon_3 = 0.1$).

When including the additional privacy budget incurred by the data-dependent approach, the data-dependent approach loses its competitive edge over the data-independent approach. Note that setting $\epsilon_2 = \epsilon_3 = 0$ would reduce the data-dependent approach to the data-independent one. The real advantage of the data-dependent approach can be best seen by allotting only a small portion of the overall privacy budget to Algorithm 1; then we can release $\hat{\theta}^P$ and $\overline{\epsilon_1^P}(\cdot)$ with reasonable overhead while achieving tighter and more accurate upper bounds for $\overline{\mu^P}(\cdot)$ and $\overline{g^P}(\cdot)$. By suffering a small additional *ex-post* pDP loss (Figure 3a), we can release the *ex-post* pDP losses of Algorithm 1 much more accurately (Figure **??**). The downside to this is that reducing $\epsilon_1$ reduces the accuracy of the output $\hat{\theta}^P$. Deciding how to allocate the privacy budget between $\epsilon_1, \epsilon_2$ and $\epsilon_3$ thus requires weighing the importance of an accurate $\hat{\theta}^P$ against the importance of an accurate $\overline{\epsilon_1^P}(\cdot)$.

## 5 Conclusion

We precisely calculate the privacy loss that an individual $z$ suffers *after* the objective perturbation algorithm is run on a specific dataset. The *ex-post* pDP loss function in DP-ERM can be accurately released to all individuals with little or no additional privacy cost. In particular, we present a *data-independent* bound which empirically provides a reasonably accurate estimate of the *ex-post* pDP loss while requiring no further privatization step. Reserving some of the privacy budget allows us to alternatively release a tighter *data-dependent* bound.

An important next step is to extend the per-instance DP analysis to the setting of constrained optimization. There are many promising future directions including using publishable pDP losses for designing more data-adaptive DP algorithms.

**Acknowledgments**

The work was partially supported by NSF CAREER Award # 2048091, Google Research Scholar Award and a gift from Evidation Health.

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
