**Part**

# Appendix

## Table of Contents

# A    DP Variants

Algorithm design is a typical use case for differential privacy: given a privacy budget of $\epsilon$, the data curator would like to add noise calibrated to meet the privacy demands. Our work concerns the converse problem of how to calculate and report the *incurred* privacy loss to an individual after a randomized algorithm is run on a fixed dataset. The table below summarizes the relevant variations of the DP definition which characterize the privacy loss with varying degrees of granularity.

Let $P, Q$ be distributions over $\Omega$, taking $p(\omega)$ and $q(\omega)$ to be the probability density/mass function of each at $\omega$. Then the probability metrics used in the table are defined as follows:

- $D_\infty(P\|Q) = \sup\limits_{S \subset \Omega} \left( \log \dfrac{P(S)}{Q(S)} \right)$   (max divergence)

- $D_\infty^\delta(P\|Q) = \sup\limits_{S \subset \Omega : P(\omega) \geq \delta} \left( \log \dfrac{P(S) - \delta}{Q(S)} \right)$   ($\delta$-approximate max divergence),

- $D_\alpha(P\|Q) = \dfrac{1}{\alpha - 1} \log \mathbb{E}_{\omega \sim Q} \left[ \left( \dfrac{p(\omega)}{q(\omega)} \right)^\alpha \right]$   (Rényi divergence).

| | |
|---|---|
| **Pure DP** | $\sup\limits_{D} \sup\limits_{z, D' : D' \simeq_z D} D_\infty\big(\mathcal{A}(D)\|\mathcal{A}(D')\big) \leq \epsilon$ |
| **Approximate DP** | $\sup\limits_{D} \sup\limits_{z, D' : D' \simeq_z D} D_\infty^\delta\big(\mathcal{A}(D)\|\mathcal{A}(D')\big) \leq \epsilon$ |
| **Rényi DP** | $\sup\limits_{D} \sup\limits_{z, D' : D' \simeq_z D} D_\alpha\big(\mathcal{A}(D)\|\mathcal{A}(D')\big) \leq \epsilon$ |
| **Data-dependent DP** | $\sup\limits_{z, D' : D' \simeq_z D} D_\alpha\big(\mathcal{A}(D)\|\mathcal{A}(D')\big) \leq \epsilon(D)$ |
| **Personalized DP** | $\sup\limits_{D, D' : D' \simeq_z D} \max\left( D_\infty^\delta\big(\mathcal{A}(D)\|\mathcal{A}(D')\big), D_\infty^\delta\big(\mathcal{A}(D')\|\mathcal{A}(D)\big) \right) \leq \epsilon(z)$ |
| **Per-instance DP** | $\max\left( D_\infty^\delta\big(\mathcal{A}(D)\|\mathcal{A}(D')\big), D_\infty^\delta\big(\mathcal{A}(D')\|\mathcal{A}(D)\big) \right) \leq \epsilon(D, z)$ |
| *Ex-post* **per-instance DP** | $\left\| \log \dfrac{\Pr\big[\mathcal{A}(D) = o\big]}{\Pr\big[\mathcal{A}(D') = o\big]} \right\| \leq \epsilon(o, D, D')$   where $D' \simeq_z D$ |

# B   Additional Experiments

## B.1   Varying dimension and dataset size

Our first experiment uses a synthetic dataset for logistic regression as described in the experiments section of the main paper. Figure 4 illustrates how the worst-case pDP loss over all individuals in the dataset – i.e., $\max_{z \in D} \epsilon_1(\hat{\theta}^P, D, D_{\pm z})$ – changes as a function of the dataset size (number of individuals in the dataset) $n$, compared to the worst-case pDP bounds given by the data-independent and data-dependent approaches. We fix $d = 50$ and vary $n$ from $n = 100$ to $n = 10000$.

Figure 4 illustrates how the worst-case pDP loss and bounds change as a function of the data dimension $d$. We fix $n = 1000$ and vary $d$ from $d = 1$ to $d = 60$. Figures 4 and 5 demonstrate that for GLMs, the strength of our *ex-post* pDP bounds $\epsilon_1^P(\cdot)$ does not depend on the size of the dataset or the dimensionality of the data.

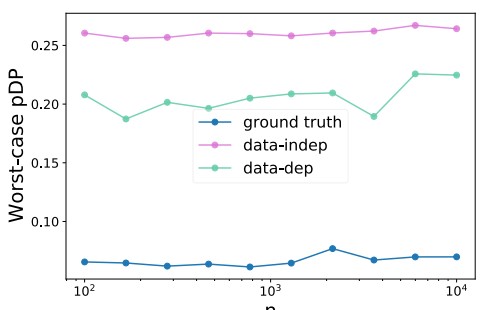

Figure 4: Worst-case pDP while varying $n$.

Figure 5: Worst-case pDP while varying $d$.

## B.2   Privacy budget allocation

Here we investigate how to distribute the privacy budget between the components of Algorithm 1 and Algorithm 2, with the same experimental setup as in Section 4.2. As before, we use the UCI credit default dataset. Our experiments show that a careful allocation of the privacy budget is essential to reaping the benefits of the data-dependent approach to releasing the *ex-post* pDP losses.

The plots in Figure 6 are ordered by increasing $\epsilon_1^{DEP}$. $\epsilon_1^{INDEP} = 1$ is fixed, as are (implicitly) $\epsilon_2^{INDEP} = \epsilon_3^{INDEP} = 0$. We see that as $\epsilon_1^{DEP}$ approaches the total privacy budget of $\epsilon_1^{INDEP} = 1$, leaving less budget for $\epsilon_2^{DEP}$ and $\epsilon_3^{DEP}$, the data-dependent release is little better than the data-independent release – worse, even, because we've expended additional privacy cost without significantly boosting the accuracy of the release.

Deciding between the data-independent or data-dependent approach is a delicate choice which depends on the particular problem setting. However, based on our theoretical and experimental results we can offer some loose guidelines:

- For non-GLMs, the data-independent bound has a dimension dependence. Therefore in the high-dimensional case, we recommend the data-dependent approach for generic convex loss functions and the data-independent approach for GLMs.

- For GLMs, the data-independent approach gives tight bounds without any overhead. The only reason to use the data-dependent approach for GLMs would be to gain an even more accurate estimate of the *ex-post* pDP losses, in which case it would be necessary to either suffer an additional privacy cost, or maintain the privacy cost by suffering a less accurate estimate of $\hat{\theta}^P$.

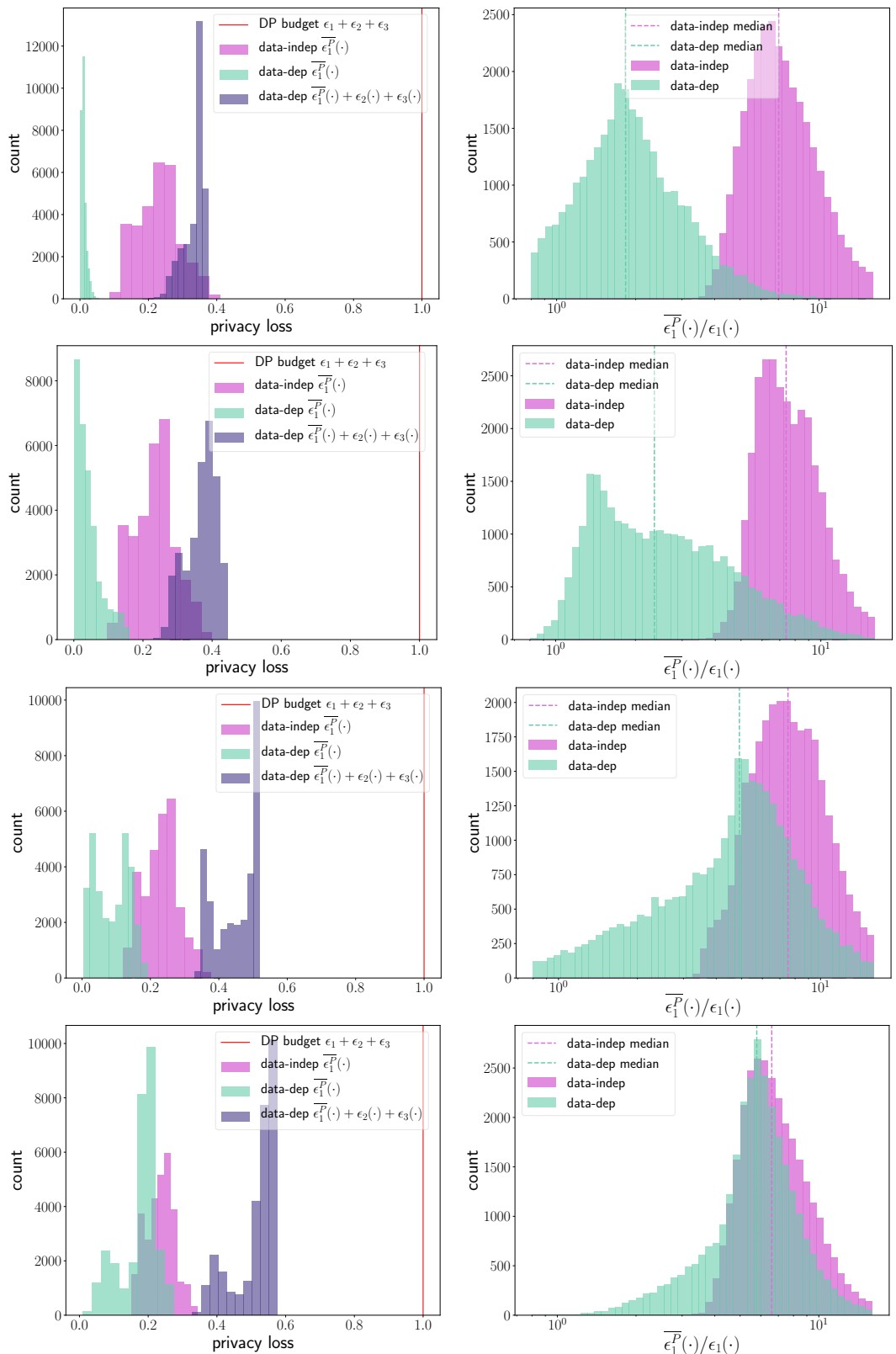

Figure 6: Data-independent release uses a privacy budget $\epsilon_1 = 1$ for each plot. From top to bottom, the budgets for the data-dependent release are $\epsilon_1 = 0.2, \epsilon_2 = 0.7, \epsilon_3 = 0.1$; $\epsilon_1 = 0.4, \epsilon_2 = 0.5, \epsilon_3 = 0.1$; $\epsilon_1 = 0.5, \epsilon_2 = 0.25, \epsilon_3 = 0.25$; and $\epsilon_1 = 0.8, \epsilon_2 = 0.1, \epsilon_3 = 0.1$.

## B.3    Comparison of pDP losses and private upper bounds

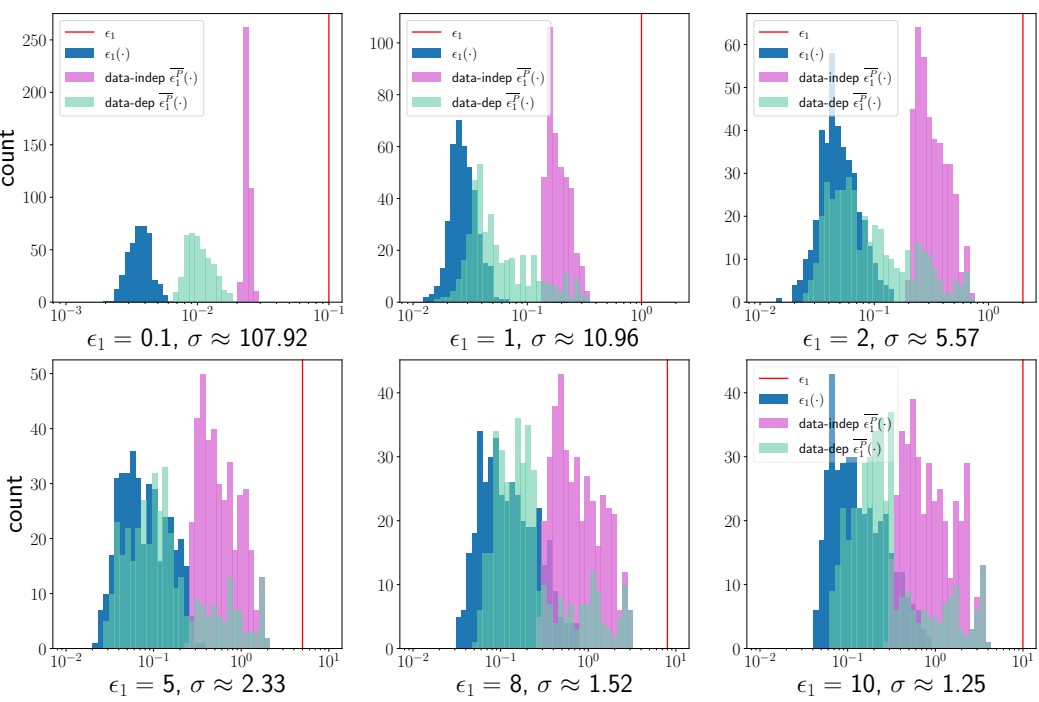

Figure 7: pDP losses $\epsilon_1(\cdot)$ and upper bounds $\overline{\epsilon_1^P}(\cdot)$ for private logistic regression applied to the UCI kidney dataset. DP budget for releasing $\hat{\theta}^P$ is $\epsilon = 1$, marked in red.

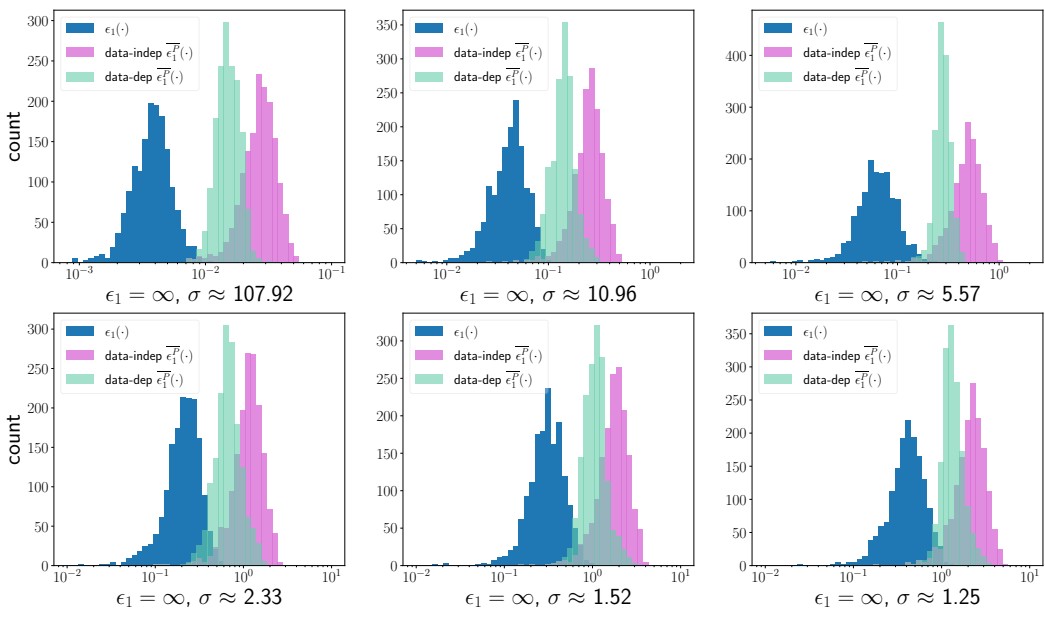

Figure 8: pDP losses $\epsilon_1(\cdot)$ and upper bounds $\overline{\epsilon_1^P}(\cdot)$ for private linear regression applied to the UCI wine quality dataset. Since we are dealing with an unbounded domain $\mathbb{R}^d$, the algorithm does not satisfy worst-case DP for any $\epsilon < \infty$.

We run both the data-independent and -dependent variations of Algorithm 1 as described in the experimental setup. Note that in this experiment the additional DP budget for the data-dependent release is $\epsilon_2 = \epsilon_3 = 1$, i.e. the privacy budget for the data-dependent release is three times the DP budget for the data-independent release. Figures 7 and 8 compare the pDP losses $\epsilon_1(\cdot)$ and private upper bounds $\overline{\epsilon_1^P}$ with $\epsilon_1$ (indicated by the vertical red line), the DP budget for Algorithm 1. Figure 7 shows results for private logistic regression on the UCI kidney dataset; Figure 8 shows results for private linear regression on the UCI wine quality dataset (Dua & Graff, 2017). Our experimental results indicate that for smaller $\epsilon_1 << 1$ (larger $\sigma$), the data-dependent approach provides a markedly tighter bound on $\epsilon_1()$·.

Figures 9 and 10 plot the ratio of the private upper bound $\epsilon_1^P(\cdot)$ for both the data-independent and -dependent approaches to the true pDP loss $\epsilon_1(\cdot)$. This illustrates the relative accuracy of the pDP estimates $\epsilon_1^P(\cdot)$. For both logistic regression on the UCI kidney dataset (Figure 9) and linear regression on the UCI wine quality dataset (Figure 10), the data-dependent approach provides a more accurate estimate of the pDP loss $\epsilon_1(\cdot)$, especially for logistic regression on the kidney dataset.

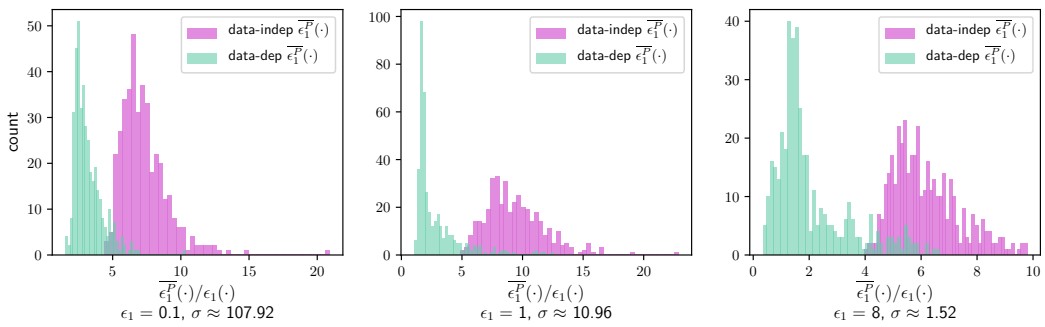

Figure 9: Ratio of private upper bound $\overline{\epsilon_1^P}(\cdot)$ to actual pDP loss $\epsilon_1(\cdot)$ for private logistic regression applied to the UCI kidney dataset.

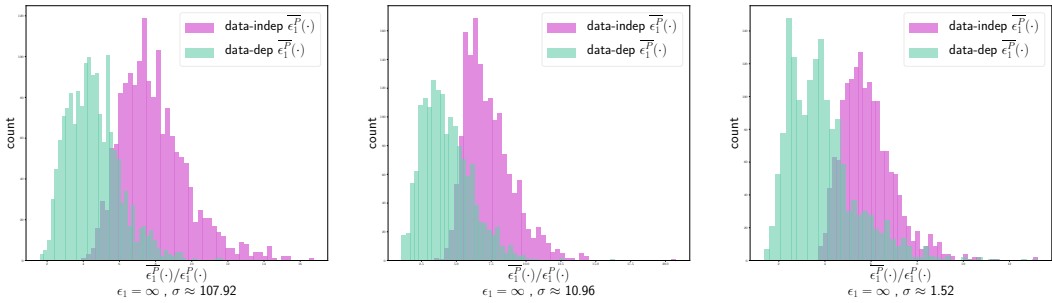

Figure 10: Ratio of private upper bound $\overline{\epsilon_1^P}(\cdot)$ to actual pDP loss $\epsilon_1(\cdot)$ for private linear regression applied to the UCI wine quality dataset.

## C Even Stronger Privacy Report

### C.1 More Accurate Privacy Report by Adapting to the Data

We now present a more adaptive version of Algorithm 2 that could be even more accurate depending on the intrinsic stability of the dataset itself. The key technical components include:

- Adapting to a well-conditioned $H$ by releasing $\lambda_{\min}$.

- A "regularized" construction of $\hat{\mu}^p(\cdot)$ that provides valid upper bounds of $\mu(\cdot)$ for all choices of $\lambda > 0$.

Algorithm 4 makes use of a subroutine to add noise to the smallest eigenvalue of $H$, presented below along with its privacy guarantees.

---

**Algorithm 3** Releasing the smallest eigenvalue of $H$

---

**Input:** Dataset $D$, noise parameter $\sigma_4$, $\lambda_{\min}$ denoting the smallest eigenvalue of $H$.
**Output:** $\hat{\lambda}^P_{\min}$.
Output $\hat{\lambda}^P_{\min} = \lambda_{\min} + \mathcal{N}(0, \sigma_4^2)$ .

---

**Theorem 11.** *Algorithm 3 satisfies pDP with*

$$\epsilon_4(\cdot) = \frac{f''(\cdot)^2 \|x\|^4}{2\sigma_4^2} + \frac{f''(\cdot)\|x\|^2 \sqrt{2\log(1/\delta)}}{\sigma_4},$$

*and if $f''(\cdot)\|x\|^2 \leq \beta$ for all $x$ then Algorithm 3 also satisfies $(\epsilon, \delta)$-DP with $\epsilon = \frac{\beta^2}{2\sigma_4^2} + \frac{\beta\sqrt{2\log(1/\delta)}}{\sigma_4}$.*

*Proof.* Algorithm 3 is a standard Gaussian mechanism. By Weyl's lemma, the smallest singular value satisfies a perturbation bound of $f''(\hat{\theta}^p; z)\|xx^T\|_2 = f''(\hat{\theta}^p; z)\|x\|^2$ from adding or removing one individual data point. The stated result follows from the theorem of the Gaussian mechanism with per-instance (and global) sensitivity set as the above perturbation bound. □

In the more general smooth-loss case we can simply replace $f''(\hat{\theta}^p; z)\|x\|^2$ with $\|\nabla^2 \ell(\hat{\theta}^p; z)\|_F$.

---

**Algorithm 4** More adaptive privacy report for `Obj-Pert`

---

**Input:** $\hat{\theta}^p$ from `Obj-Pert`, noise parameter $\sigma, \sigma_2, \sigma_3, \sigma_4$; regularization parameter $\lambda$; Hessian $H := \sum_i \nabla^2 \ell(\hat{\theta}^p; z_i) + \lambda I_d$, failure probability $\rho$.

**Output:** Reporting function $\tilde{\epsilon} : (x, y), \rho \to \mathbb{R}_+^2$.

Privately release $\hat{g}^p$ by Algorithm 5 with parameter $\sigma_2$.

Set $\epsilon_2(\cdot)$ according to Theorem 14.

Set $\overline{g^P}(z) := f'(\cdot)[\hat{g}^P]^T x + \sigma_2 \|f'(\cdot)x\|_2 F_{\mathcal{N}(0,1)}^{-1}(1 - \rho/2)$.

Set $\tau = F_{\lambda_1(\mathrm{GOE}(d))}^{-1}(1 - \rho/2)$.

Privately release $\hat{H}^p$ by Algorithm 6 with parameter $\sigma_3$.

Set $\epsilon_3(\cdot)$ according to Theorem 17

Privately release $\hat{\lambda}_{\min}^p = \lambda_{\min} + \mathcal{N}(0, \sigma_4^2)$ (Algorithm 3).

Set $\epsilon_4(\cdot)$ according to Theorem 11.

Set $\underline{\hat{\lambda}_{\min}^p} := \max\{\lambda, \hat{\lambda}_{\min}^p - \sigma_4 F_{\mathcal{N}(0,1)}^{-1}(1 - \rho/2)\}$.

**if** $\underline{\hat{\lambda}_{\min}^p} \geq 2\tau\sigma_3$ **then**

  Set $\overline{\mu^p}(x) = \min\left\{\frac{\underline{\hat{\lambda}_{\min}^p} + \tau\sigma_3}{\underline{\hat{\lambda}_{\min}^p}} x^T(\hat{H}^P)^{-1}x, \frac{\|x\|^2}{\underline{\hat{\lambda}_{\min}^p}}\right\}$. (use the standard estimator)

**else**

  Set $\overline{\mu^p}(x) = \min\left\{\frac{\underline{\hat{\lambda}_{\min}^p} + 2\tau\sigma_3}{\underline{\hat{\lambda}_{\min}^p}} x^T(\hat{H}^p + \tau\sigma_3 I_d)^{-1}x, \frac{\|x\|^2}{\underline{\hat{\lambda}_{\min}^p}}\right\}$. (use the regularized estimator)

**end if**

Set $\overline{\epsilon_1^p}(\cdot) := \left| -\log\left(1 - f''(\cdot)\overline{\mu^P}(x)\right) \right| + \frac{\|f'(\cdot)x\|_2^2}{2\sigma^2} + \frac{|\overline{g^P}(z)|}{\sigma^2}$.

Return the "privacy report" function $\tilde{\epsilon} = (\overline{\epsilon_1^p}, \epsilon_2 + \epsilon_3 + \epsilon_4)$, i.e., the *ex-post* pDP of Algorithm 1 and the pDP of Algorithm 4 (i.e., overhead).

---

This algorithm allows any choice of $\lambda$ to be used in ObjPert, so that the privacy report is non-intrusive and can be attached to an existing workflow without changing the main algorithm at all. The following proposition shows that $\overline{\mu}^p(x)$ is always a valid upper bound of the leverage score $\mu(x)$ and it is accurate if $\lambda_{\min}$ is large (from either the Hessian or the regularization).

**Proposition 12** (Uniform multiplicative approximation). *Let $\underline{\hat{\lambda}_{\min}^p}$ and $\hat{H}^P$ be constructed as in Algorithm 4. Then with probability $1 - 2\rho$,*

$$\lambda_{\min} - \sigma_4 F_{\mathcal{N}(0,1)}^{-1}(1 - \rho/2) \leq \hat{\lambda}_{\min}^p \leq \lambda_{\min} + \sigma_4 F_{\mathcal{N}(0,1)}^{-1}(1 - \rho/2)$$

*and for all $x \in \mathbb{R}^d$ simultaneously, the regularized estimator obeys that*

$$x^T(\hat{H}^P + \tau\sigma_3 I_d)^{-1}x \leq \mu(x) \leq \frac{\underline{\hat{\lambda}_{\min}^p} + 2\tau\sigma_3}{\underline{\hat{\lambda}_{\min}^p}} x^T(\hat{H}^P + \tau\sigma_3 I_d)^{-1}x.$$

*Moreover, under the same high-probability event, if $\underline{\hat{\lambda}_{\min}^p} \geq 2\tau\sigma_3$, then the standard estimator obeys that*

$$\frac{\underline{\hat{\lambda}_{\min}^p} - \tau\sigma_3}{\underline{\hat{\lambda}_{\min}^p}} x^T(\hat{H}^P)^{-1}x \leq x^T H^{-1} x \leq \frac{\underline{\hat{\lambda}_{\min}^p} + \tau\sigma_3}{\underline{\hat{\lambda}_{\min}^p}} x^T(\hat{H}^P)^{-1}x.$$

*Proof.* By Lemma 20, if we choose $\tau = F_{\lambda_1(\mathrm{GOE}(d))}^{-1}(1 - \rho/2)$, then with probability $1 - \rho$, the GOE noise matrix $G$ satisfies that $\|G\|_2 \prec \tau$, the following holds: $-\tau I_d \prec G \prec \tau I_d$.

Next, by the definition of Gaussian CDF, with probability $1 - \rho$,

$$\lambda_{\min} - \sigma_4 F_{\mathcal{N}(0,1)}^{-1}(1 - \rho/2) \leq \hat{\lambda}_{\min}^p \leq \lambda_{\min} + \sigma_4 F_{\mathcal{N}(0,1)}^{-1}(1 - \rho/2)$$

which implies that $\lambda_{\min} \geq \underline{\hat{\lambda}_{\min}^p}$, i.e.,

$$H - \underline{\hat{\lambda}_{\min}^p} I_d \succ 0$$

Therefore with probability $1 - 2\rho$,

$$H \prec H + G + \tau\sigma_3 I_d \prec H + 2\tau\sigma_3 I = H - \hat{\underline{\lambda}}^p_{\min} I_d + \hat{\underline{\lambda}}^p_{\min} I_d + 2\tau\sigma_3 I_d$$

$$\prec \frac{\hat{\underline{\lambda}}^p_{\min} + 2\tau\sigma_3}{\hat{\underline{\lambda}}^p_{\min}}(H - \hat{\underline{\lambda}}^p_{\min} I_d + \hat{\underline{\lambda}}^p_{\min} I_d) = \frac{\hat{\underline{\lambda}}^p_{\min} + 2\tau\sigma_3}{\hat{\underline{\lambda}}^p_{\min}} H,$$

where the first semidefinite inequality uses that $H - \hat{\underline{\lambda}}^p_{\min} I_d$ is positive semi-definite.

Taking the inverse on both sides, we get

$$\frac{\hat{\underline{\lambda}}^p_{\min}}{\hat{\underline{\lambda}}^p_{\min} + 2\tau\sigma_3} H^{-1} \prec (\hat{H}^P + \tau\sigma_3 I_d)^{-1} \prec H^{-1}.$$

Thus for all $x \in \mathbb{R}^d$, $x^T(\hat{H}^P + \tau\sigma_3 I_d)^{-1}x \leq x^T H^{-1} x \leq \frac{\hat{\underline{\lambda}}^p_{\min} + 2\tau\sigma_3}{\hat{\underline{\lambda}}^p_{\min}} x^T(\hat{H}^P + \tau\sigma_3 I_d)^{-1}x$, which finishes the proof for the regularized estimator.

Now we turn to the standard (unregularized) estimator. Under the same high-probability event:

$$H + G \prec H + \tau\sigma_3 I = H - \hat{\underline{\lambda}}^p_{\min} I_d + \hat{\underline{\lambda}}^p_{\min} I_d + \tau\sigma_3 I_d$$

$$\prec \frac{\hat{\underline{\lambda}}^p_{\min} + \tau\sigma_3}{\hat{\underline{\lambda}}^p_{\min}}(H - \hat{\underline{\lambda}}^p_{\min} I_d + \hat{\underline{\lambda}}^p_{\min} I_d) = \frac{\hat{\underline{\lambda}}^p_{\min} + \tau\sigma_3}{\hat{\underline{\lambda}}^p_{\min}} H.$$

Similarly,

$$H + G \succ H - \tau\sigma_3 I_d \succ H - \hat{\underline{\lambda}}^p_{\min} I_d + \hat{\underline{\lambda}}^p_{\min} I_d - \tau\sigma_3 I_d$$

$$\succ \frac{\hat{\underline{\lambda}}^p_{\min} - \tau\sigma_3}{\hat{\underline{\lambda}}^p_{\min}}(H - \hat{\underline{\lambda}}^p_{\min} I_d + \hat{\underline{\lambda}}^p_{\min} I_d) = \frac{\hat{\underline{\lambda}}^p_{\min} - \tau\sigma_3}{\hat{\underline{\lambda}}^p_{\min}} H.$$

Together the above two inequalities give

$$\frac{\hat{\underline{\lambda}}^p_{\min} - \tau\sigma_3}{\hat{\underline{\lambda}}^p_{\min}} H \prec H + G \prec \frac{\hat{\underline{\lambda}}^p_{\min} + \tau\sigma_3}{\hat{\underline{\lambda}}^p_{\min}} H.$$

Take the inverse on both sides we get

$$\frac{\hat{\underline{\lambda}}^p_{\min}}{\hat{\underline{\lambda}}^p_{\min} + \tau\sigma_3} H^{-1} \prec (\hat{H}^P)^{-1} \prec \frac{\hat{\underline{\lambda}}^p_{\min}}{\hat{\underline{\lambda}}^p_{\min} - \tau\sigma_3} H^{-1},$$

which implies that for all $x \in \mathbb{R}^d$, $\frac{\hat{\underline{\lambda}}^p_{\min} - \tau\sigma_3}{\hat{\underline{\lambda}}^p_{\min}} x^T(\hat{H}^P)^{-1}x \leq x^T H^{-1} x \leq \frac{\hat{\underline{\lambda}}^p_{\min} + \tau\sigma_3}{\hat{\underline{\lambda}}^p_{\min}} x^T(\hat{H}^P)^{-1}x$ as stated in the proposition. $\qquad\square$

The privacy (DP and pDP) of Algorithm 4 is a composition of the stated results in Theorem 10 with the the privacy guarantees stated in Theorem 11. Observe that if we choose $\sigma_3 = \sigma_1$ then the additional DP and pDP losses are smaller than those of the main algorithm, i.e., we have a constant overhead in terms of the privacy loss.

The next theorem shows that when $\lambda_{\min}(H) \to +\infty$ as the number of data points $n \to +\infty$, we could improve the leverage score part of the pDP losses from a multiplicative factor of 12 to $1 + o(1)$.

**Theorem 13** (Utility of Adaptive privacy report.). *Assume $\lambda_{\min}(H) \geq \max\{2\beta, 2\tau\sigma_3\}$. There is a universal constant $0 < C \leq 4\tau\sigma_3 + 2\beta$ such that for a fixed $z \in \mathcal{X} \times \mathcal{Y}$, and all $\rho > 0$, the privately released privacy report $\overline{\epsilon}^P_1(\cdot)$ from Algorithm 4 obeys that*

$$\epsilon_1(\cdot) \leq \overline{\epsilon}^P_1(\cdot) \leq (1 + \frac{C}{\lambda_{\min}})\epsilon_1(\cdot) + \frac{|f'(\cdot)|\|x\|}{\sigma_2}\sqrt{2\log(2/\rho)}$$

*with probability $1 - 3\rho$ where $\epsilon_1$ is the expression from Theorem 6.*

*Proof of Theorem 13.* Similar to the proof of Theorem 10, it suffices to consider the approximation of the first term when we replace $\mu$ with $\overline{\mu^p}$. First of all, by a union bound, the high probability event in Proposition 12 and the high probability event in Theorem 9 (to bound the third term in the *ex-post* pDP of ObjPert) holds simultaneously with probability at least $1 - 3\rho$. The remainder of the proof conditions on this event.

Observe that it suffices to construct a multiplicative approximation bound for the first term $\log(1 + f''(\cdot)\mu)$ or $-\log(1 - f''(\cdot)\mu)$.

By our assumption that $\lambda > 2\beta$, as well as the pointwise minimum in the construction of $\overline{\mu^p}$ from Algorithm 4, we know that $\overline{\mu^p} \le 1/2$ and $\log(1 - f''(\cdot)\overline{\mu^p})$ is well-defined.

Using the fact that for all $a \ge -1$, $\frac{a}{1+a} \le \log(1 + a) \le a$, we will now derive the multiplicative approximation for both $\log(1 + f''(\cdot)\mu)$ or $-\log(1 - f''(\cdot)\mu)$ using the plug-ins: $\log(1 + f''(\cdot)\overline{\mu^p})$ or $-\log(1 - f''(\cdot)\overline{\mu^p})$.

For brevity, in the subsequent derivation we will be using $a$ to denote $f''(\cdot)\mu(x)$ and $\hat{a}$ to denote $f''(x^T\hat{\theta}^p; y)\overline{\mu^p}(x)$.

Thus

$$\log(1 + a) \le \log(1 + \hat{a}) \le \hat{a} \le (1 + \frac{2\tau\sigma_3}{\underline{\hat{\lambda}^p_{\min}}})a \le (1 + \frac{2\tau\sigma_3}{\underline{\hat{\lambda}^p_{\min}}})(1 + a)\log(1 + a)$$
$$\le (1 + \frac{4\tau\sigma_3}{\lambda_{\min}})(1 + \frac{\beta}{\lambda_{\min}})\log(1 + a) \le (1 + \frac{C}{\lambda_{\min}})\log(1 + a)$$

where $C$ can be taken as $4\tau\sigma_3 + 2\beta$, by our assumption on $\lambda_{\min}$ and a high probability bound under which $\underline{\hat{\lambda}^p_{\min}} \ge \lambda_{\min}/2$.

Similarly,

$$-\log(1 - a) \le \frac{a}{1 - a} \le \frac{\hat{a}}{1 - a} \le \frac{(1 + \frac{2\tau\sigma_3}{\underline{\hat{\lambda}^p_{\min}}})a}{1 - a} \le \frac{(1 + \frac{2\tau\sigma_3}{\underline{\hat{\lambda}^p_{\min}}})}{1 - \frac{\beta}{\lambda_{\min}}}(-\log(1 - a))$$

where

$$\frac{(1 + \frac{2\tau\sigma_3}{\underline{\hat{\lambda}^p_{\min}}})}{1 - \frac{\beta}{\lambda_{\min}}} = 1 + \frac{2\tau\sigma_3}{\underline{\hat{\lambda}^p_{\min}}} + \frac{\beta/\lambda_{\min}}{1 - \beta/\lambda_{\min}} \le 1 + \frac{4\tau\sigma_3 + 2\beta}{\lambda_{\min}}$$

under our assumption for $\lambda_{\min}, \beta$. The additive error term in the third term follows from the same bound as in the non-adaptive result without any changes.

The version for the standard (non-regularized) version is similar and is left as an exercise. $\square$

### C.2   Dataset-Dependent Privacy report for general smooth learning problems

So far, we have focused on generalized linear losses. Most of our results can be extended to general smooth learning problems.

For the third term in the pDP bound of Theorem 10, the challenge is that the two vectors are now nontrivially coupled with each other via $\hat{\theta}^p$. For this reason we propose to privately release the gradient at $\hat{\theta}^p$, which helps to decouple the dependence and allow a tighter approximation at a small cost of accuracy and additional privacy budget.

For convenience, we will denote $g = \nabla J(\hat{\theta}^P; D)^T \nabla \ell(\hat{\theta}^P; z)$. Below, we present an algorithm that outputs $g^P$ (a private approximation of $g$) as well as the additional privacy cost $\epsilon_4(\cdot)$ of outputting $g^P$.

---

**Algorithm 5** Release $g^P$, a private approximation of $g = \nabla J(\hat{\theta}^P; D)^T \nabla \ell(\hat{\theta}^P; z)$

---

**Input:** Dataset $D$, privatized output $\hat{\theta}^P$, noise parameter $\sigma_2$, linear loss function $L(\theta; D) = \sum_i \ell(\theta; z_i)$, regularization parameter $\lambda$, convex and twice-differentiable regularizer $r$.
**Output:** $g^P(\cdot), \epsilon_2(\cdot)$.
Construct noise vector $e \sim \mathcal{N}(0, \sigma_2^2 I)$.
Set $J^P := \nabla L(\hat{\theta}^P; D) + \nabla r(\theta) + \lambda \hat{\theta}^P + e$.
Set $g^P(\cdot)$ s.t. $g^P(z) = (J^P)^T \nabla_z \ell(\hat{\theta}^P; z)$.
Set $\epsilon_2(\cdot)$ s.t. $\epsilon_2(z) = \frac{\|\nabla \ell(\hat{\theta}^P; z)\|^2}{2\sigma_2^2} + \frac{\|\nabla \ell(\hat{\theta}^P; z)\|\sqrt{2\log(2/\delta)}}{\sigma_2}$.

---

**Theorem 14.** *Let $\hat{\theta}^P$ be fixed, Algorithm 5 satisfies*

1. $(\epsilon_2(D, D_{\pm z}), \delta)$-*pDP, with*

$$\epsilon_2(D, D_{\pm z}) = \frac{\|\nabla \ell(\hat{\theta}^P; z)\|^2}{2\sigma_2^2} + \frac{\|\nabla \ell(\hat{\theta}^P; z)\|\sqrt{2\log(1/\delta)}}{\sigma_2}.$$

2. $\epsilon_2(o, D, D_{\pm z})$-*ex post pDP with probability $1 - \rho$,*

$$\epsilon_2(o, D, D_{\pm z}) = \frac{\|\nabla \ell(\hat{\theta}^P; z)\|^2}{2\sigma_2^2} + \frac{\|\nabla \ell(\hat{\theta}^P; z)\|\sqrt{2\log(2/\rho)}}{\sigma_2}.$$

*Proof.* This is a Gaussian mechanism and the proof follows from Corollary 23. $\qquad \square$

The theorem avoids an additional dependence in $d$ from the $\ell_1$-norm $\|\nabla \ell(\hat{\theta}^p; z)\|_1$ in the dataset-independent bound.

We remark that Algorithm 5's pDP loss is dataset-independent and if we choose $\sigma_2 = \sigma_1$, the pDP losses for running Algorithm 5 are on the same order as those of the main algorithm. Thus the additional overhead is on the same order and no recursive privacy reporting is needed.

For the first term, our release of $H$ and $\lambda_{\min}$ extends without any changes to the more general case. The estimator of the leverage score needs to be modified accordingly.

We defer the analysis of how accurately this estimator approximates the first term of $\epsilon_1(\cdot)$ to a longer version of the paper.

### C.3  Uniform Privacy Report and Privacy Calibration

The "privacy report" algorithm (Algorithm 2) that we presented in the main paper and the "adaptive privacy report" (Algorithm 4 is straightforward and omitted. focus on releasing a reporting function $\tilde{\epsilon}$ that is accurate with high probability for every fixed input.

Sometimes there is a need to ensure that with high probability, $\tilde{\epsilon}$ is accurate *simultaneously* for all $z_1, ..., z_n$ in the dataset, or even for all $z \in \mathcal{Z}$ for a data domain $\mathcal{Z}$. The following theorem shows that this is possible at a mild additional cost in the accuracy. These results are stated for Algorithm 2), but extensions to that of Algorithm 4 is straightforward and thus omitted.

**Proposition 15** (Uniform privacy report). *With probability $1 - 2\rho$, simultaneously for all $n$ users in the dataset, the output of Algorithm 2 obeys that $\epsilon_1(\hat{\theta}^p, D, D_{\pm z}) \leq \overline{\epsilon_1^P}(\hat{\theta}^p, z) \leq 12\epsilon_1(\hat{\theta}^p, D, D_{\pm z}) + \frac{|f'(\cdot)|\|x\|}{\sigma_2}\sqrt{2\log(n/\rho)}$.*

*If we, instead, use the data-independent bound $\frac{|f'(x^T\hat{\theta}^p; y)|\|x\|_1\sqrt{2\log(2d/\rho)}}{\sigma}$ to replace the third-term in $\overline{\epsilon_1^P}(\cdot)$, then with probability $1 - 2\rho$, simultaneously for all $x \in \mathcal{X}$, the ex-post pDP report $\overline{\epsilon_1^P}$ from Algorithm 2 satisfies that*

$$\epsilon_1(\cdot) \leq \overline{\epsilon_1^P}(z, \hat{\theta}^p) \leq 12\epsilon_1(\cdot) + \frac{|f'(\cdot)|\|x\|_1\sqrt{2\log(2d/\rho)}}{\sigma}.$$

*Proof.* We note that the approximation of $\mu_x$ is uniform for all $x$. It remains to consider a uniform bound for the third term over the randomness of ObjPert. The first statement follows by taking a union bound. The second result is achieved by Holder's inequality, the concentration of max of i.i.d. Gaussians. □

Sometimes it is desirable to calibrate the noise-level to a prescribed "worst-case" DP parameter $\epsilon, \delta$. The following corollary explains that the additional DP loss and pDP losses when we calibrate Algorithm 2 with the same privacy parameter as those in Algorithm 1 will yield a total DP and pDP that are at most twice as large under an additional condition that $f'' \leq f'$.

**Corollary 16** (The additional privacy cost). *If we calibrate $\sigma_2$ such that the Algorithm 2 satisfies the same $(\epsilon, \delta)$-DP as Algorithm 1, i.e., when $\epsilon < 1$, we could choose $\sigma_2 = \frac{\rho_{\max}}{\epsilon}\sqrt{2\log(1.25/\delta)}$. Then Algorithm 2 satisfies $(\epsilon(\cdot), \delta)$-pDP with*

$$\epsilon(\cdot) = \frac{\epsilon^2 (f''(\cdot))^2 \|x\|^4}{8\rho_{\max}^2 \log(1.25/\delta)} + \frac{\epsilon(f''(\cdot))\|x\|^2}{\rho_{\max}\sqrt{2}}.$$

*For those cases when $\frac{(f''(\cdot))\|x\|^2}{\rho_{\max}} \leq \frac{|f'(\cdot)|\|x\|}{\beta}$ (which is the case in logistic regression for all $x$ s.t., $\|x\| \leq 1$), the additional overhead in releasing a dataset-dependent pDP is smaller than the ex post pDP bound in Theorem 6.*

## D    Improved "Analyze Gauss" with Gaussian Orthogonal Ensembles

In this section we propose a differentially private mechanism that releases a matrix $H$ when

$$H = \sum_{i=1}^{n} H_x$$

where $H_x \in \mathbb{R}^{d \times d}$ is a symmetric matrix computed from individual data point $x$.

Examples of this include

1.  (unnormalized / uncentered) sample covariance $H_x = xx^T$
2.  Empirical Fisher information $H_x = \nabla\ell(\theta; x)\nabla\ell(\theta; x)T$ where $\ell$ is the log-likelihood and $\theta$ is the true parameter;
3.  Hessian of a generalized linear loss function $H_x = f''(x, \theta)xx^T$.
4.  Hessian of a smooth loss function $H_x = \nabla^2\ell(x, \theta)$.

In the first three cases $H_x$ is a rank-1 matrix and our use case in this paper is the third and fourth example. Throughout this section we assume $\|H_x\|_F \leq \beta$ for all $x \in \mathcal{X}$.

The mechanism we propose is a variant of "Analyze-Gauss" (Dwork et al., 2014b) but it reduces the required variance of the added noise by a factor of 2 in almost all coordinates hence resulting in higher utility.

The standard "Analyze-Gauss" leverages the symmetry of $H$ and uses the standard Gaussian mechanism to release the upper triangular region (including the diagonal) of the matrix $H$ with an $\ell_2$-sensitivity upper bound:

$$\|\text{UpperTriangle}(H) - \text{UpperTriangle}(H')\|_2 \leq \|H_x\|_F \leq \beta.$$

where $\text{UpperTriangle}(H) \in \mathbb{R}^{d^2/2+d/2}$ is the vector that enumerates the elements of the upper-triangular region of $H$. The resulting Gaussian noise is distributed i.i.d as $\mathcal{N}(0, \sigma_3^2)$ and it satisfies $(\epsilon, \delta)$-DP with

$$\epsilon = \frac{\beta^2}{2\sigma_3^2} + \frac{\beta\sqrt{2\log(1/\delta)}}{\sigma_3}.$$

The alternative that we propose also adds a symmetric noise but doubles the variance on the diagonal elements.

---

**Algorithm 6** Release $H$ (a natural variant of "Analyze-Gauss")

---

**Input:** Dataset $D$, noise parameter $\sigma_3$, $H = \sum_{i=1}^{n} \nabla^2 \ell(z_i, \hat{\theta}^P) + \lambda I_d$.
**Output:** $\hat{H}^P$.
Draw a Gaussian random matrix $Z \in \mathbb{R}^{d \times d}$ with $Z_{i,j} \sim \mathcal{N}(0, \sigma_3^2)$ independently.
Output $\hat{H}^P = H + \frac{1}{\sqrt{2}}(Z + Z^T)$.

---

The symmetric random matrix $\frac{1}{\sqrt{2}}(Z + Z^T)$ is known as the Gaussian Orthogonal Ensemble (GOE) and well-studied in the random matrix theory. We will first show this this mechanism obeys DP and pDP.

**Theorem 17.** *Algorithm 6 satisfies pDP with*

$$\epsilon(\cdot) = \frac{\|H_x\|_F^2}{4\sigma_3^2} + \frac{\|H_x\|_F \sqrt{2\log(1/\delta)}}{\sqrt{2}\sigma_3},$$

*and $\hat{H}^p$ satisfies ex post pDP of the same $\epsilon$ with probability $1 - 2\delta$. If in addition $\sup_{x \in \mathcal{X}} \|H_x\|_F \leq \beta$ then, Algorithm 6 satisfies $(\epsilon, \delta)$-DP with*

$$\epsilon \leq \frac{\beta^2}{4\sigma_3^2} + \frac{\beta\sqrt{2\log(1/\delta)}}{\sqrt{2}\sigma_3}.$$

**Improvements over "Analyze Gauss".** Notice that if we choose $\sigma_3$ to be $1/\sqrt{2}$ of the noise scale with used in the standard "Analyze Gauss", we will be adding the same amount of noise on the diagonal, achieve the same DP and pDP bounds, while adding noise with only half the variance in the off-diagonal elements. The idea is to add noise with respect to the natural geometry of the sensitivity, as we illustrate in the proof.

*Proof.* Algorithm 6 is equivalent to releasing the vector $[f_1, f_2]$ using a standard Gaussian mechanism with $\mathcal{N}(0, \sigma_3^2 I_{\frac{d^2}{2} + d/2})$, where $f_1 \in \mathbb{R}^d$ is the diagonal of $H/\sqrt{2}$ and $f_2 \in \mathbb{R}^{(d^2-d)/2}$ is the vectorized the strict upper triangular part of $H$.

The per-instance $\ell_2$-sensitivity of $[f_1, f_2]$ is

$$\|\Delta_x\|_2 = \sqrt{\sum_{1 \leq i < j \leq d} H_x[i,j]^2 + \sum_{k=1^d} H_x[k,k]^2 (1/\sqrt{2})^2}$$

$$= \sqrt{\frac{1}{2}\left(\sum_{1 \leq i < j \leq d} H_x[i,j]^2 + \sum_{1 \leq j < i \leq d} H_x[i,j]^2 + \sum_{k=1^d} H_x[k,k]^2\right)}$$

$$= \frac{1}{\sqrt{2}}\|H_x\|_F$$

The result then follows from an application of the pDP computation of the Gaussian mechanism. $\quad\square$

### D.1   Exact statistical inference with the Gaussian Orthogonal Ensemble

Besides a constant improvement in the required noise, another major advantage of using the Gaussian Orthogonal Ensemble is that we know the exact distribution of its eigenvalues (Chiani, 2014) which makes statistical inference, e.g., constructing confidence intervals, easy and constant-tight.

**Lemma 18** (Largest singular value of Gaussian random matrix (Rudelson & Vershynin, 2010, Equation (2.4))). *Let $A \in \mathbb{R}^{d \times d}$ be a random matrix with i.i.d. $\sigma^2$-subgaussian entries, then there exists universal constants $C, c$ such that for all $t > 0$*

$$\mathbb{P}[s_{\max}(A) \geq (2+t)\sqrt{d\sigma^2}] \leq Ce^{-cdt^{3/2}}.$$

*i.e., with probability $1 - \delta$*

$$\|A\|_2 \leq \left(2 + \left(\frac{(\log(C/\delta))}{cd}\right)^{2/3}\right)\sqrt{d\sigma^2}.$$

Notice that the symmetric matrix, i.e., Gaussian orthogonal ensemble is identically distributed to $\frac{1}{\sqrt{2}}(Z + Z^T)$ where $Z$ is a iid Gaussian random matrix, thus by triangular inequality, we have

**Lemma 19** (Largest eigenvalue of Gaussian orthogonal ensemble). *Let $A$ be a Gaussian orthogonal ensemble (i.e., a symmetric random matrix with $\mathcal{N}(0, \sigma^2)$ on the off-diagonal and $\mathcal{N}(0, 2\sigma^2)$ on the diagonal), with probability $1 - \delta$,*

$$\|A\|_2 \leq \sqrt{2}\left(2 + \left(\frac{(\log(C/\delta))}{cd}\right)^{2/3}\right)\sqrt{d\sigma^2}.$$

*Proof.* The proof follows from triangular inequality of the spectral norm. $\qquad\square$

The above bound is asymptotic and we will use it for deriving the theoretical results. For practical computation, the the exact formula of the CDF of the largest eigenvalue of GOE matrices is given by (Chiani, 2014, Theorem 2). We could use this to bound the spectral norm of the noise added to Algorithm 6.

**Lemma 20.** *Let $A$ be described as in Lemma 19.*

$$\|A\|_2 \leq \sigma F_{\lambda_1 \text{ of GOE}}^{-1}(1 - \rho/2)$$

*where $F_{\lambda_1 \text{ of GOE}}$ is the CDF of the largest eigenvalue of the standard GOE matrix with constructed by $\frac{1}{\sqrt{2}}(Z + Z^T)$ where each element of matrix $Z$ is drawn i.i.d. from a standard gaussian.*

*Proof.* Notice that the GOE matrix is symmetric, so the largest eigenvalue $\lambda_1$ and the negative of the smallest eigenvalue $-\lambda_d$ are identically distributed. Thus the operator norm $\|A\|_2 \leq \max\{|\lambda_1|, |\lambda_d|\} \leq F_{\lambda_1 \text{ of GOE}}^{-1}(1 - \rho/2)$ with probability $1 - \rho$. $\qquad\square$

**Numerical computation:** Chiani (2014, Theorem 2) characterized the distribution of $\lambda_1$ and provided an exact analytical formula with stable numerical implementation to compute $F_{\lambda_1 \text{ of GOE}}$. Thus $F_{\lambda_1 \text{ of GOE}}^{-1}$ can be evaluated using a binary search.

Using the Mathematica implementation provided by (Chiani, 2014), we find that $F_{\lambda_1 \text{ of GOE(50)}}^{-1}(1 - \rho/2) = 12$ for $\rho = 8.465 \times 10^{-6}$. Therefore in our experiments with $d = 50$, we choose $\tau \approx 12$.

## E  Omitted Proofs

With the two technical components presented, we are now ready to present the detailed proofs of our main results: Theorem 6 and Theorem 10.

### E.1  Proofs for the pDP analysis of objective perturbation

*Proof of Theorem 6.* We calculate the *ex-post* pDP loss of Algorithm 1 as follows. Consider the perturbed objective function:

$$\hat{\theta}^P = \underset{\theta \in \mathbb{R}^d}{\arg\min} \hat{\mathcal{L}}(\theta; D) + r(\theta) + \frac{\lambda}{2}\|\theta\|_2^2 + b^T\theta. \tag{2}$$

Let $\mathcal{A}$ be the algorithm that outputs $\hat{\theta}^P$ as stated in 2. The *ex-post* per-instance privacy loss (with the abuse of notation discussed in Section 2.1) is then given by

$$\epsilon_1(\theta, D, D_{\pm z}) = \max\left(\log \frac{\Pr(\mathcal{A}(D) = \hat{\theta}^P)}{\Pr(\mathcal{A}(D_{\pm z}) = \hat{\theta}^P)}, \log \frac{\Pr(\mathcal{A}(D_{\pm z}) = \hat{\theta}^P)}{\Pr(\mathcal{A}(D) = \hat{\theta}^P)}\right),$$

Note that this characterization of *ex-post* per-instance DP is equivalent to that stated in Definition 3, since switching the numerator and denominator of a log ratio is the same as flipping its sign.

Since we can't easily calculate the distribution of $\hat{\theta}^P$, we will instead use the bijection between the output $\hat{\theta}^P$ and the noise vector $b$ (observed in Chaudhuri et al. (2011)) to rewrite the log probability ratio more cleanly.

First-order conditions applied to (2) tell us that

$$b(\hat{\theta}^P; D) = -\left( \nabla \hat{\mathcal{L}}(\hat{\theta}^P; D) + \nabla r(\hat{\theta}^P) + \lambda \hat{\theta}^P \right). \tag{3}$$

Then taking the gradient of the noise vector, we have

$$\nabla b(\hat{\theta}^P; D) = -\left( \nabla^2 \hat{\mathcal{L}}(\hat{\theta}^P; D) + \nabla^2 r(\hat{\theta}^P) + \lambda I_d \right). \tag{4}$$

Let $b \sim \mathcal{N}(0, \sigma^2 I_d)$, and denote $\nu(\cdot)$ as the probability density function of the normal distribution: i.e., the density at $b$ is $\nu(b; \sigma) \propto e^{-\frac{||b||_2^2}{2\sigma^2}}$. Then since the objective function $J(\theta; D)$ is strictly convex in $\theta$ (implying as in Chaudhuri et al. (2011) that the mapping between $\hat{\theta}^P$ and $b$ is bijective and monotonic), by Lemma 30 we can write

$$
\begin{aligned}
\log \frac{\Pr(\mathcal{A}(D) = \hat{\theta}^P)}{\Pr(\mathcal{A}(D_{\pm z}) = \hat{\theta}^P)} &= \log \frac{\left| \det\left( \nabla b(\hat{\theta}^P; D) \right) \right|}{\left| \det\left( \nabla b(\hat{\theta}^P; D_{\pm z}) \right) \right|} \frac{\nu(b(\hat{\theta}^P; D); \sigma)}{\nu(b(\hat{\theta}^P; D_{\pm z}); \sigma)} \\
&= \log \frac{\left| \det\left( \nabla b(\hat{\theta}^P; D) \right) \right|}{\left| \det\left( \nabla b(\hat{\theta}^P; D_{\pm z}) \right) \right|} + \log \frac{e^{-\frac{1}{2\sigma^2} ||b(\hat{\theta}^P; D)||_2^2}}{e^{-\frac{1}{2\sigma^2} ||b(\hat{\theta}^P; D_{\pm z})||_2^2}} \\
&= \underbrace{\log \frac{\left| \det\left( \nabla b(\hat{\theta}^P; D) \right) \right|}{\left| \det\left( \nabla b(\hat{\theta}^P; D_{\pm z}) \right) \right|}}_{(*)} + \underbrace{\frac{1}{2\sigma^2} \left( ||b(\hat{\theta}^P; D_{\pm z})||_2^2 - ||b(\hat{\theta}^P; D)||_2^2 \right)}_{(**)}.
\end{aligned}
$$

Dealing first with the term (*), we observe that $\nabla b(\hat{\theta}^P; D_{\pm z}) = \nabla b(\hat{\theta}^P; D) \mp \nabla^2 \ell(\hat{\theta}^P; z)$. The notation "$\mp$" means to subtract if $z \notin D$, and add if $z \in D$. Using the eigendecomposition $\nabla^2 \ell(\hat{\theta}^P; z) = \sum_{k=1}^d \lambda_k u_k u_k^T$ and recursively applying the matrix determinant lemma, we have

$$
\begin{aligned}
\left| \det\left( \nabla b(\hat{\theta}^P; D_{\pm z}) \right) \right| &= \left| \det\left( \nabla b(\hat{\theta}^P; D) \mp \nabla^2 \ell(\hat{\theta}^P; z) \right) \right| \\
&= \left| \det\left( \nabla b(\hat{\theta}^P; D) \mp \sum_{k=1}^d \lambda_k u_k u_k^T \right) \right| \\
&= \left| \det\left( \nabla b(\hat{\theta}^P; D) \mp \sum_{k=1}^{d-1} \lambda_k u_k u_k^T \mp \lambda_d u_d u_d^T \right) \right| \\
&= \left| \det\left( \nabla b(\hat{\theta}^P; D) \mp \sum_{k=1}^{d-1} \lambda_k u_k u_k^T \right) \right| \left( 1 \mp \lambda_d u_d^T \left( \nabla b(\hat{\theta}^P; D) \mp \sum_{k=1}^{d-1} \lambda_k u_k u_k^T \right)^{-1} u_d \right) \\
&= \ldots \\
&= \left| \det\left( \nabla b(\hat{\theta}^P; D) \right) \right| \prod_{j=1}^d \left( 1 \mp \mu_j \right),
\end{aligned}
$$

where $\mu_j = \lambda_j u_j^T \left( \nabla b(\hat{\theta}^P; D) \mp \sum_{k=1}^{j-1} \lambda_k u_k u_k^T \right)^{-1} u_j$. Therefore,

$$(*) = \log \frac{\left|\det\big(\nabla b(\hat{\theta}^P; D)\big)\right|}{\left|\det\big(\nabla b(\hat{\theta}^P; D_{\pm z})\big)\right|}$$

$$= \log \frac{\left|\det\big(\nabla b(\hat{\theta}^P; D)\big)\right|}{\left|\det\Big(\nabla b(\hat{\theta}^P; D)\Big)\right| \prod_{j=1}^{d} \big(1 \mp \mu_j\big)}$$

$$= \log \frac{1}{\prod_{j=1}^{d} \big(1 \mp \mu_j\big)}$$

$$= -\log \prod_{j=1}^{d} \big(1 \mp \mu_j\big).$$

We'll handle the second term (**) next. We have that

$$(**) = \frac{1}{2\sigma^2}\Big(||b(\hat{\theta}^P; D_{\pm z})||_2^2 - ||b(\hat{\theta}^P; D)||_2^2\Big)$$

$$= \frac{1}{2\sigma^2}\Big[\mp \nabla \ell(\hat{\theta}^P; z)\Big]\Big[2b(\hat{\theta}^P; D) \mp \nabla \ell(\hat{\theta}^P; z)\Big]$$

$$= \pm \frac{1}{\sigma^2}\big[\nabla J(\hat{\theta}^P; D)^T \nabla \ell(\hat{\theta}^P; z)\big] + \frac{1}{2\sigma^2}||\nabla \ell(\hat{\theta}^P; z)||_2^2.$$

The rest of the proof follows from adding together (*) and (**), and taking the absolute value. $\qquad\square$

*Proof of Corollary 7.* By restricting to generalized linear models, we can give a more interpretable pDP result for Algorithm 1 with the main difference being a cleaner version of the generalized leverage score. In the case of GLMs, we have that $\nabla \ell(\cdot) = f'(\cdot)x$ and $\nabla^2 \ell(\cdot) = f''(\cdot)xx^T$. So $\ell(\cdot)$ has a rank-one Hessian with only one eigenvalue, and $\log \prod_{j=1}^{d}\big(1 + \mu_j\big) = \log(1 + \mu(x))$. Here $\mu_j$ is as defined in Theorem 6 and $\mu$ is defined as in Corollary 7. $\qquad\square$

*Proof of Theorem 8.* Using the eigendecomposition $\nabla^2 \ell(\hat{\theta}^P; z) = \sum_{k=1}^{d} \lambda_k u_k u_k^T$, for $0 \leq j \leq d$ we have that

$$\mu_j(x) = \begin{cases} \lambda_j u_j^T \Big( -\nabla^2 L(\hat{\theta}^P; D) - \lambda I_d - \nabla^2 r(\hat{\theta}^P) - \sum_{k=1}^{j-1} \lambda_k u_k u_k^T \Big)^{-1} u_j & \text{if } z \notin D \\ \lambda_j u_j^T \Big( -\sum_{\substack{z_i \in D \\ z_i \neq z}} \nabla^2 \ell(\hat{\theta}^P; z_i) - \lambda I_d - \nabla^2 r(\hat{\theta}^P) - \sum_{k=j}^{d} \lambda_k u_k u_k^T \Big)^{-1} u_j & \text{if } z \in D. \end{cases}$$

$$:= \begin{cases} \lambda_j u_j^T H_{+z}^{-1} u_j & \text{if } z \notin D \\ \lambda_j u_j^T H_{-z}^{-1} u_j & \text{if } z \in D. \end{cases}$$

The second equality introduces the shorthand $\mu_j(x) := \lambda_j u_j^T H_{\pm z}^{-1} u_j$. Observe that $\nabla^2 \ell(\hat{\theta}^P; z_i), \nabla^2 r(\hat{\theta}^P) \in \mathbb{R}^{d \times d}$ are positive semi-definite, since $\ell(\cdot)$ and $r(\theta)$ by assumption are convex functions with continuous second-order partial derivatives. Since $\nabla^2 \ell(\hat{\theta}^P; z_i)$ is PSD, its eigenvalues are non-negative and so $\lambda_k \geq 0$ for all $0 \leq k \leq d$. Then for any $x \in \mathbb{R}^d$, $x^T u_k u_k^T x = (x^T u_k)^2 \geq 0$. So $u_k u_k^T$ is also PSD, and we then have that $H_{+z} + \lambda I_d$ and $H_{-z} + \lambda I_d$ are both negative semi-definite. Therefore, $H_{\pm z} \prec -\lambda I_d$ and after taking the inverse, we see that $\mu_j(x) \leq -\frac{\lambda_j}{\lambda} \leq 0$ or equivalently $-\mu_j(x) \geq \frac{\lambda_j}{\lambda} \geq 0$.

For $-1 < \mu_j(x) \le 0$, we have that

$$
\begin{aligned}
\big| -\log(1 - \mu_j(x)) \big| &= \log(1 + (-\mu_j(x))) \\
&\le -\mu_j(x) \\
&\le -\log(1 + \mu_j(x)) \\
&= \big| -\log(1 + \mu_j(x)) \big| \\
&\le -\log(1 - \frac{\lambda_j}{\lambda}).
\end{aligned}
$$

The rest of the proof follows from converting the log-product into a sum of logs. For a linear loss function $\ell(\theta; z) = f(x^T\theta; y)$, the simplified bound can be achieved due to the rank-one Hessian $\nabla^2\ell(\hat{\theta}^P; z) = f''(x^T\hat{\theta}^P; y)xx^T$ whose only eigenvalue is $\lambda_1 = f''(x^T\hat{\theta}^P; y)\|x\|_2^2$. $\qquad\square$

*Proof of Theorem 9.* By Holder's inequality,

$$
\left| \nabla J(\hat{\theta}^P; D)\nabla\ell(\hat{\theta}^P; z) \right| \le ||\nabla J(\hat{\theta}^P; D)||_\infty ||\ell(\hat{\theta}^P; z)||_1.
$$

Recall from (3) that $\nabla J(\hat{\theta}^P; D) = -b(\hat{\theta}^P; D)$. Therefore $||\nabla J(\hat{\theta}^P)||_\infty = \max_{i \in [d]} |b_i|$, where $b_i \sim \mathcal{N}(0, \sigma^2)$. Applying a union bound and using the standard Gaussian tail bound,

$$
\begin{aligned}
\Pr\left[ \max_{i \in [d]} |b_i| \ge t \right] &= \Pr\left[ \bigcup_i |b_i| \ge t \right] \\
&\le \sum_{i \in [d]} \Pr\big[ |b_i| \ge t \big] \\
&\le 2d e^{-\frac{t^2}{2\sigma^2}}.
\end{aligned}
$$

So with probability $1 - \rho$, we have $||\nabla^J(\hat{\theta}^P; D)||_\infty \le \sigma\sqrt{2\log(2d/\rho)}$. The stronger bound for linear loss functions comes from substituting $||\nabla\ell(\hat{\theta}^P)||_1 = f'(x^T\theta; y)||x||_1$. $\qquad\square$

### E.2   Proofs for the Privacy Report in the main paper

The proof of Theorem 10 relies on the following intermediate result.

**Proposition 21** (Uniform multiplicative approximation). *If $\lambda_{\min}(H) \ge 2\sigma_2 F^{-1}_{\lambda_1(\mathrm{GOE}(d))}(1 - \rho/2)$, then with probability $1 - \rho$, for all $x \in \mathbb{R}^d$ simultaneously*

$$
\frac{1}{2}x^T(\hat{H}^P)^{-1}x \le x^T H^{-1}x \le \frac{3}{2}x^T(\hat{H}^P)^{-1}x.
$$

*Proof.* By the choice of $\tau = F^{-1}_{\lambda_1(\mathrm{GOE}(d))}(1 - \rho/2)$, with probability $1 - \rho$, the noise matrix $Z$ from the release of $\hat{H}^P$ satisfies that $\|Z\|_2 \le \sigma_2\tau \le \lambda_{\min}/2$. Thus $-\frac{H}{2} \prec -\frac{\lambda_{\min}}{2}I_d \prec Z \prec \frac{\lambda_{\min}}{2}I_d \prec \frac{H}{2}$. Adding $H$ on both sides

$$
\frac{H}{2} \prec H + Z \prec \frac{3H}{2}
$$

which implies that

$$
\frac{2}{3}H^{-1} \le (H + Z)^{-1} \prec 2H^{-1}.
$$

By definition of semidefinite ordering, for all $x \in \mathbb{R}^d$

$$
\frac{2}{3}x^T H^{-1}x \le x^T(H + Z)^{-1}x \le 2x^T H^{-1}x.
$$

In other word, $\frac{1}{2}\hat{\mu}_1^P(x) \le \mu_1(x) \le \frac{3}{2}\hat{\mu}_1^P(x)$. $\qquad\square$

*Proof of Theorem 10.* The privacy guarantees (Statement 1-3) follow directly from the pDP analysis in Theorem 17 that analyzes the release of $H$ by adding a GOE noise matrix and the Gaussian mechanism that releases $g$.

By the result follows from Proposition 21 we know that with probability $1 - \rho$, for all $x$

$$\mu(x) \leq \frac{3}{2}\hat{\mu}^p(x) \leq 3\mu(x)$$

For all $a \geq -1$ $\frac{a}{1+a} \leq \log(1 + a) \leq a$. Recall that $\beta \geq \sup_z \|\nabla^2 \ell(\hat{\theta}^p; z)\|2$. By our condition that $\lambda > 2\beta$, as well as the pointwise minimum in the construction of $\overline{\mu^p}$, we have that $f''\overline{\mu^p} \leq \frac{1}{2}$ and

$$\frac{f''\overline{\mu^p}}{2} \leq \max\{\log(1 + f''\overline{\mu^p}), -\log(1 - f''\overline{\mu^p}\} \leq 2f''\overline{\mu^p}.$$

Thus

$$\log(1 + f''\mu) \leq f''\mu \leq f''\overline{\mu^p} \leq 2\log(1 + f''\overline{\mu^p}) \leq 2f''\overline{\mu^p} \leq 3f''\hat{\mu}^p \leq 6f''\mu \leq 12\log(1 + f''\mu),$$

and similarly

$$-\log(1 - f''\mu) \leq 2f''\mu \leq 2\overline{f''\mu^p} \leq -2\log(1 - f''\overline{\mu^p}) \leq 4f''\overline{\mu^p} \leq 6f''\hat{\mu}^p \leq 12f''\mu \leq -12\log(1 - f''\mu).$$

This concludes the factor 12 multiplicative approximation in the first term of $\epsilon_1(\cdot)$. The second term of $\epsilon_1(\cdot)$ does not involve an approximation. The third term of $\epsilon_1(\cdot)$ is random and the bound is off by an additive factor of $\min\{\sigma, \sigma_2\}|f'(\cdot)|\|x\|_2\sqrt{2\log(2/\rho)}$ — via the smaller of the data-dependent bound and the data-independent bound, each holds with probability $1 - \rho/2$. $\quad\square$

## F  pDP Analysis of the Gaussian mechanism

**Theorem 22** (*ex-post* pDP of Gaussian mechanism). *Let $Q : \mathcal{Z}^* \to \mathbb{R}^d$ be a function of the data. Let $|Q(D_{\pm z}) - Q(D)| \leq \Delta_z$. Then the Gaussian mechanism that releases $o \sim Q(D) + \mathcal{N}(0, \sigma^2 I_d)$ obeys* ex-post *pDP with*

$$\epsilon(o, D, D_z) = \left| \frac{\|\Delta_z\|^2}{2\sigma^2} - \frac{\Delta_z^T(o - Q(D))}{\sigma^2} \right|.$$

*Proof.* We can directly calculate the log-odds ratio:

$$\frac{1}{2\sigma^2} \left( \|o - Q(D)\|^2 - \|o - Q(D_{\pm z})\|^2 \right)$$
$$= \frac{1}{2\sigma^2} \left( (Q(D_{\pm z}) - Q(D))^T(2o - Q(D) - Q(D_{\pm z})) \right)$$
$$= \frac{1}{2\sigma^2} \left( \Delta_z^T(2o - 2Q(D) - \Delta_z) \right)$$
$$= \frac{-\|\Delta_z\|^2}{2\sigma^2} + \frac{\Delta_z^T(o - Q(D))}{\sigma^2}.$$

The proof is complete by taking the absolute value. $\quad\square$

**Corollary 23** (pDP bound and high-probability ex-post pDP of Gaussian mechanism). *Let $\Phi$ be the cumulative distribution function (CDF) of a standard normal random variable. The Gaussian mechanism that releases $o \sim Q(D) + \mathcal{N}(0, \sigma^2 I_d)$ satisfies dataset independent pDP bound with*

$$\epsilon(D, D_{\pm z}) \leq \frac{\|\Delta_z\|^2}{2\sigma^2} + \frac{\|\Delta_z\|\Phi^{-1}(1 - \delta)}{\sigma} \leq \frac{\|\Delta_z\|^2}{2\sigma^2} + \frac{\|\Delta_z\|\Phi^{-1}(1 - \delta)}{\sigma}.$$

*Moreoever, with probability at least $1 - \rho$ over the distribution of the randomized output $o$, the Gaussian mechanism satisfies obeys the following dataset-independent* ex post *pDP bound*

$$\epsilon(o, D, D_{\pm z}) \leq \frac{\|\Delta_z\|^2}{2\sigma^2} + \frac{\|\Delta_z\|\Phi^{-1}(1 - \rho/2)}{\sigma} \leq \frac{\|\Delta_z\|^2}{2\sigma^2} + \frac{\|\Delta_z\|\sqrt{2\log(2/\rho)}}{\sigma}. \quad (5)$$

*Proof.* Since $o \sim Q(D) + \mathcal{N}(0, \sigma^2 I_d)$, we have $\Delta_z^T(o - Q(D)) \sim \mathcal{N}(0, \sigma^2 \|\Delta_z\|^2)$. The results of pDP follows from the tailbound of the privacy loss random variable and Lemma 27.

For the high-probability bound of the *ex post* pDP, we need to bound both sides of the privacy loss random variable. It suffices to show that the absolute value of the added noise is bounded with a union bound on the two-sided tails, each with probability $1 - \rho/2$. $\qquad\square$

A tighter pDP bound can be obtained using the analytical Gaussian mechanism (Balle & Wang, 2018). We choose to present the tail bound-based formula above for the interpretability of the results.

# G   Technical Lemmas

**Lemma 24** (Sherman-Morrison-Woodbury Formula). *Let $A, U, C, V$ be matrices of compatible size. Assuming $A, C$ and $C^{-1} + V A^{-1} U$ are all invertible, then*

$$(A + UCV)^{-1} = A^{-1} - A^{-1}U(C^{-1} + VA^{-1}U)^{-1}VA^{-1}.$$

**Lemma 25** (Determinant of Rank-1 perturbation). *For invertible matrix $A$ and vector $c, d$ of compatible dimension*

$$\det(A + cd^T) = \det(A)(1 + d^T A^{-1} c).$$

**Lemma 26** (Gaussian tail bound). *Let $X \sim \mathcal{N}(0, \sigma^2)$. Then*

$$\mathbb{P}(X > \sigma\epsilon) \leq \frac{e^{-\epsilon^2/2}}{\epsilon}.$$

*A convenient alternative representation (slightly weaker) is*

$$\mathbb{P}(X > \sigma\sqrt{2\log(1/\delta)}) \leq \delta,$$

*and*

$$\mathbb{P}(|X| > \sigma\sqrt{2\log(2/\delta)}) \leq \delta.$$

*for all $\delta > 0$.*

**Lemma 27** (Tail bound to $(\epsilon, \delta)$-DP conversion). *Let $\epsilon(o) = \log(\frac{p(o)}{p'(o)})$ where $p$ and $p'$ are densities of $\theta$. If*

$$\mathbb{P}_p(\epsilon(o) > \epsilon) \leq \delta$$

*then for any measurable set $\mathcal{S}$*

$$\mathbb{P}_p(\theta \in \mathcal{S}) \leq e^\epsilon \mathbb{P}_{p'}(\theta \in \mathcal{S}) + \delta.$$

*Two useful applications of this result for DP are:*

1. *if $\mathbb{P}_p(\epsilon(o) > \epsilon) \leq \delta$ for all pairs of neighboring dataset $D, D'$ such that $p = \mathcal{A}(D), p' = \mathcal{A}(D')$ then $\mathcal{A}$ is $(\epsilon, \delta)$-DP.*

2. *If $D' = D_{\pm z}$, $p = \mathcal{A}(D), p' = \mathcal{A}(D_{\pm z})$ and that $\mathbb{P}_p(\epsilon(o) > \epsilon) \leq \delta$ and $\mathbb{P}_{p'}(-\epsilon(o) < -\epsilon) \leq \delta$, then $\mathcal{A}$ satisfies $(\epsilon, \delta)$-pDP for individual $z$ and dataset $D$.*

*Proof.* Let $E$ be the event that $|\epsilon(\theta)| > t$, by definition it implies that for any $\tilde{E} \subset E$, $\mathbb{P}_p(\theta \in \tilde{E}) \leq e^t \mathbb{P}_{p'}(\theta \in \tilde{E})$. Now consider any measurable set $\mathcal{S}$:

$$\mathbb{P}_p(\theta \in \mathcal{S}) = \mathbb{P}_p(\theta \in \mathcal{S} \cap E^c) + \mathbb{P}_p(\theta \in \mathcal{S} \cap E)$$
$$\leq \mathbb{P}_{p'}(\theta \in \mathcal{S} \cap E^c)e^t + \mathbb{P}_p(\theta \in E) \leq e^t \mathbb{P}_{p'}(\theta \in \mathcal{S}) + \delta.$$

The two applications follow directly from the definitions of $(\epsilon, \delta)$-DP and pDP. $\qquad\square$

**Lemma 28** (maximum of subgaussian). *Let $X_1, ..., X_n$ be iid $\sigma^2$-subgaussian random variables.*

$$\mathbb{P}[\max_i X_i \geq \sqrt{2\sigma^2(\log n + t)}] \leq e^{-t}.$$

*Proof.* The proof is by standard subgaussian concentration and union bound. $\qquad\square$

**Lemma 29** (Weyl's theorem; Theorem 4.11, p. 204 in Stewart (1990))**.** *. Let $A, E$ be given $m \times n$ matrices with $m \geq n$, then*

$$\max_{i \in [n]} |\sigma_i(A) - \sigma_i(A + E)| \leq \|E\|_2 \tag{6}$$

**Lemma 30** ("Change-of-variables" for density functions)**.** *Let $g : \mathbb{R}^d \to \mathbb{R}^d$ be a bijective and differentiable function, and let $X, Y$ be continuous random variables in $\mathbb{R}^d$ related by the transformation $Y = g(X)$. Then the probability density of $Y$ is*

$$f_Y(y) = f_X(g^{-1}(y)) \left| det \left[ \frac{\partial g^{-1}(y)}{\partial y} \right] \right|,$$

*with $\left[ \dfrac{\partial g^{-1}(y)}{\partial y} \right]$ denoting the $d \times d$ Jacobian matrix of the mapping $X = g^{-1}(Y)$.*

**Lemma 31.** (Billboard lemma) *Suppose $\mathcal{A} : \mathcal{D} \to \mathcal{R}$ satisfies $(\epsilon, \delta)$ differential privacy. Consider any set of functions $f_i : \mathcal{D}_i \times \mathcal{R} \to \mathcal{R}$, where $\mathcal{D}_i$ is the portion of the dataset containing individual $i$'s data. The composition $\{f_i(\Pi_i D, \mathcal{A}(D))\}$ satisfies $(\epsilon, \delta)$-joint differential privacy, where $\Pi_i : \mathcal{D} \to \mathcal{D}_i$ is the projection to individual $i$'s data.*