# OpenReview forum: "Privately Publishable Per-instance Privacy"
_NeurIPS.cc/2021/Conference — NeurIPS 2021 Poster_

### Official Review · Reviewer_fBsx · 2021-07-06

**Rating:** 7
**Confidence:** 3

**Summary:**

The paper looks at how to release per-instance differentially private (pDP) privacy bounds, focusing on empirical risk minimization (ERM) with a convex loss function. The paper proposes methods for releasing either a looser data independent estimate or a tighter data-dependent one. Without properly checking the proofs, the paper seems sensible and interesting.

**Limitations And Societal Impact:**

There is not much discussion on the limitations, although the limiting assumptions are stated clearly enough. I do not think this paper needs a separate discussion on societal impact.

**Main Review:**

** update after rebuttals & discussion
I lower my score a bit due to concerns raised by other reviewers (see the other reviews for more specific points). However, I still think this is a nice paper and ready to be published.
**

The paper focuses on pDP, basically a datapoint-specific measure of privacy, for ERM with convex losses using objective perturbation for privacy. With pDP the privacy guarantees depend on the sensitive data, and therefore cannot be released without some further privacy mechanism. The authors propose methods for releasing the pDP privacy values: a looser bound holds independent of the actul data, while under some additional assumptions a tighter data-dependent bound can be used.
The paper is mostly clear and easy to read. In short, I have some questions for the authors, but overall the introduced methods seem novel and interesting.

Questions/comments for the authors:
1) It is unclear to me how the different epsilon values relate to each other, i.e., based on the paper I am not sure if plotting epsilons from standard DP vs pDP makes sense (are they on the same scale, are the privacy guarantees somehow comparable when the epsilons are equal, or how should I interpret the difference in epsilons). Can you elaborate a bit?

Minor comments:
* Definitions 1 & 2 using input space of fixed size n seems a bit weird considering the neighbourhood relation.
* I especially liked the connection to existing statistical theory pointed out in the form generalized leverage score.

**Time Spent Reviewing:**

5

---

> ### Author Response · Authors · 2021-08-10
> **Author Response**
>
> Thanks for your thoughtful review! We’ve tried to address your comments and questions below.
>
> **>>> It is unclear to me how the different epsilon values relate to each other, i.e., based on the paper I am not sure if plotting epsilons from standard DP vs pDP makes sense (are they on the same scale, are the privacy guarantees somehow comparable when the epsilons are equal, or how should I interpret the difference in epsilons). Can you elaborate a bit?**
>
> DP and pDP are on the same scale and their privacy guarantees can be interpreted identically when they have the same value if we consider a particular individual $z$ and a dataset $D$ being taken as an input.  The difference is that DP’s $\epsilon$ is a scalar that applies to all individuals and all possible inputs, while pDP’s $\epsilon(\cdot)$ is a function that differs for each individual and each input dataset. We refer to the original paper [Wang, 2019] for a more detailed discussion on the semantics of pDP and its properties.
>
> A slightly subtle difference in this paper is that we also considered *ex-post* pDP, which shares the same semantics as *ex-post* DP. *Ex-pos*t DP differs from (*ex-ante*) DP in that it depends on the realized random (and differentially private) output, resulting in a more fine-grained value of $\epsilon$. They are still on the same scale.  A (not rigorous, but somewhat telling) analogy is that: “*ex-post* pDP is to pDP what a realized random variable is to the same R.V.’s high-probability bound.”
>
>
> **>>> Minor comments:
> Definitions 1 & 2 using input space of fixed size n seems a bit weird considering the neighbourhood relation.**
>
> Thank you for spotting the inconsistency!  We are using $\mathcal{Z}^* = \cup_{n=0}^{\infty} \mathcal{Z}^n$ to denote the space of datasets (with an unspecified number of data points) and the randomized algorithm $\mathcal{A}$ should be defined as one that takes any element of $\mathcal{Z}^*$ rather than $\mathcal{Z}^n$ for a fixed $n$. We will correct this in Definition 1 and 2.  Please verify that ObjPert and all other DP mechanisms we considered in this paper are indeed valid under this correct definition.

---

### Official Review · Reviewer_NN5N · 2021-07-09

**Rating:** 6
**Confidence:** 4

**Summary:**

The paper builds upon a recent work of Wang that introduced per-instance differential privacy, a refinement of the usual approximate differential privacy which captures the effect of a specific data point in the context of a specific dataset on a DP analysis. In particular, the paper proposes ways of reporting per-instance DP losses in a privacy-preserving manner (satisfying joint DP, essentially), focusing on convex ERM with objective perturbation.

**Limitations And Societal Impact:**

Some limitations are addressed in the conclusion section. The paper makes a contribution to the theory of differential privacy, which as a general research topic is of great societal relevance. I do not see potential negative societal impacts.

**Main Review:**

The paper has two main contributions: the first one is evaluating per-instance privacy losses for objective perturbation, and the second one is designing mechanisms for reporting such losses. The motivation for making such reports lies in the fact that the overall DP parameter can be a conservative overestimate of the per-instance privacy loss. While the paper does a good job of emphasizing this gap between per-instance and overall DP, as well as explaining that per-instance parameters are sensitive, I think it could be more convincing in describing why we would want to make these reports in the first place. It’s certainly nice to be able to report these values to participants, but it felt like a bit of a niche objective without more motivation.

In terms of the results, I have various specific comments and questions. The presentation is not completely clear at times; some notation is never defined and I had trouble understanding certain statements.

1. The data-independent bound on pDP is just a bound on personalized DP of Ebadi, Sands, Schneider (2015), right? You’re not really doing anything “per-instance” there? Note that personalized DP immediately satisfies joint DP.
2. Related to the previous point, in the experiments you only plot the data-independent bound. Can you demonstrate empirically that it makes sense to go through the effort of providing a data-dependent bound? Again, for me per-instance differential privacy means that you’re data-dependent; otherwise you’re reporting personalized DP.
3. I didn’t understand Figure 1 and its setup. In the lines starting with 213, there is a setup where I don’t see any randomization (denoted by b previously) in how the solution is computed. I don’t understand how what’s done there satisfies DP, what the purpose of the rotation is, and how the figure relates to Corollary 7. I might be missing something obvious.
4. In statements such as Theorem 6, what do you mean by $\pm$? You mean whichever sign would maximize the expression?
5. In line 223, what do you mean by “hidden in this analysis are the delta’s of Algorithm 1”? I don’t see any delta in Algorithm 1.
6. What is the function b(…) in Theorem 6 and Theorem 8? It’s never defined.
7. What is $F^{-1}_{\lambda(GOE(d))}$ in Algorithm 2? I don’t think it’s defined anywhere.
8. I’m not sure the $D_{\pm z}$ notation works. In Definition 3, for example, say D doesn’t contain z. Then, you’re only bounding one direction of the divergence, when $\mathcal{A}(D)$ is in the numerator and $\mathcal{A}(D_{\pm z})$ in the denominator. You’re not bounding the other direction.
9. In line 161, why does it say “simplified version”? I don’t see the difference between Algorithm 1 and equation (1).
10. I don’t understand the expression “privacy target z” in Theorem 6/Corollary 7.
11. I don’t understand the comment in line 168 that says “chooses a regularization strength to ensure that the objective function is strongly convex”. The objective in equation (1) is strongly convex for any non-zero lambda since all other functions are convex as well.
12. In line 290, is theta $N(0, \mathbb{I})$ (it’s not 1-d)?
13. In line 291, what do you mean by Y being a deterministic function of X and theta such that …? I’m confused by several things here. First, how is Y deterministic if you’re defining Pr[Y=1]? Second, how is the probability equal to something + noise matrix? What are the dimensions of E, X, theta here?
14. Could you clarify what result you’re referring to in the Bassily et al reference in line 281?
15. In line 155, you could clarify which norm you’re considering (e.g., $\ell_2$).
16. In line 275, is it supposed to be $\tilde \epsilon$ between $\epsilon_1$ and $12\epsilon_1$?

------------------------------------------------------------------------------------------------

Thank you for your response. It was indeed very helpful in clarifying various technical specifics and notation, so I'm happy to increase my score. That said, I'm still a bit confused about the motivation. I don't see why joint DP is the right requirement if we want to handle the "accuracy first" setup or justifying large global epsilons (in addition, as you described the accuracy first setup it sounds like there would be privacy leakage through the noise level, which your results might not handle). It's a nice feature to have the possibility of making these joint DP reports, but perhaps I still see it as a bit of a niche objective. I can see how building on these results you might be able to improve data-dependent approaches like propose-test-release, as you suggested, and I think this is a very promising direction and I'm eager to read these results. Unfortunately, the current submission does not give any indication of how this could be done, which is why I'm hesitant to increase my score further.

**Time Spent Reviewing:**

7

---

> ### Author Response · Authors · 2021-08-10
> **Author Response**
>
> Thanks for the very detailed feedback, and the time you took to review the paper! We’ve done our best to address your comments and questions.
>
> As for motivating why we would want to make these reports in the first place:  In most use cases, accuracy is more important than privacy. This is the same practical assumption that motivated Ligett et al’s “Accuracy First” paper, in which the data analyst wishes to maximize privacy given a fixed accuracy constraint. Per-instance DP could be valuable in a similar setting where $\epsilon$ is determined according to a desired level of utility, with the key difference being that we propose to adaptively analyze the privacy loss to particular individuals after the computation ends.
>
> Notice that it has recently become more common for researchers to use privacy parameters as high as $\epsilon = 8$ or $\epsilon=16$ in empirical DP work, and then justify that these seemingly pointless privacy guarantees are actually much stronger in practice by conducting membership inference attacks (see, e.g., https://arxiv.org/abs/2009.10031 and https://arxiv.org/abs/2106.09352).  Publishing pDP losses provides a better and theoretically sound alternative to the empirical approach for revealing the gap between the worst-case DP bound and the actual privacy loss in practice.
>
> We are also considering how pDP could be useful in designing data-dependent DP algorithms, e.g. combining the framework with a “propose-test-release” approach.
>
> Below are our responses to the other comments and questions.
>
> **1.** The data-independent bound on pDP is indeed closely related to personalized DP. However, Ebadi et al don’t look at objective perturbation, nor do they consider an *ex-post* setting. Both the data-independent and -dependent bounds are implicitly functions of $\hat{\theta}^P$ through $f’(\hat{\theta}^P; z)$ and $f''(\hat{\theta}^P; z)$. The parameter $\hat{\theta}^P$ is drawn from a distribution that depends on the fixed dataset *D*, and so the choice of an *ex-post* setting --in which the pDP loss is a function of the algorithm’s output $\hat{\theta}^P$ -- distinguishes the data-independent pDP bound from personalized DP.
>
>
> **2.** We’ve updated the experiments to compare the data-independent and -dependent bound. The other modification we’ve made is to the privacy report: now, the data-dependent approach adds noise not just to the Hessian (to release the first term of the pDP loss $\epsilon_1(\cdot)$ of Algorithm 1), but also to the gradient (to release the third term of the pDP loss). In our updated experiments with linear and logistic regression on several datasets, the data-dependent approach gives a tighter bound on Algorithm 1’s pDP losses, but the data-independent approach is a more practical choice after factoring in the additional pDP costs of releasing $\overline{\epsilon_1^P}(\cdot)$ via the data-dependent approach.
>
> We speculate that this might not be the case for non-GLMs, for which releasing the third term of $\epsilon_1(\cdot)$ via a data-independent bound would have a necessary dependence on dimension *d* (as illustrated in Theorem 9). Then we would expect to see that — especially for higher-dimensional data — the data-dependent bound is tighter even when including the additional privacy costs of adding noise to the Hessian and the gradient. Furthermore, the data-dependent bound on the first term of $\epsilon_1(\cdot)$ would be tighter when the Hessian is well-conditioned, compared to the data-independent release of the first term which uses a more conservative bound based only on the value of the regularization parameter $\lambda$.
>
>
> **3.** The aim of Figure 1 is to provide a visualization of the *ex-post* pDP losses by directly plugging in different values of $\hat{\theta}^P$, where $\hat{\theta}^P$ is determined via a rotation of the non-private classifier $\hat{\theta}$ rather than by the coin flips of Algorithm 1. The color of each data point in Figure 1 corresponds to its *ex-post* pDP value as calculated according to Corollary 7. Figure 1 is supposed to demonstrate how the value of $\hat{\theta}^P$ mediates between the first term of $\epsilon_1(\cdot)$ dominating (for data points that are influential in fitting the model) and the third term dominating (for data points that are mis-classified).
>
> Note that $\hat{\theta}^P$ is random due to ObjPert and it satisfies DP; *ex-post* pDP calculates the incurred privacy loss after the random $\hat{\theta}^P$ is realized (therefore ”*ex-post*").
>
>
> **4.** We use “$\pm$” to mean “add if $z \notin D$, subtract otherwise”. Similarly. "$\mp$” means “subtract if $z \notin D$, add otherwise”. The description of this notation should have been included in Section 2.1.
>
> The exception to this is that in the expression for $\overline{\epsilon_1^P}(\cdot)$ in Algorithm 2,  “$\pm$” does mean whichever sign would maximize the expression.
>
>
> **5.** It might have been clearer to write “the delta’s of Theorem 5” rather than “the delta’s of Algorithm 1”. What was meant by that is that the *ex-ante* privacy guarantees of Obj-Pert stated in Theorem 5 are contingent upon a certain setting of $\lambda, \sigma, \delta$ in order to achieve ($\epsilon, \delta$)-DP. Using *ex-post* privacy, however, does away with the $\delta$’s -- the coins have already been flipped, the output released and we need no longer consider the entire probability space of the randomized algorithm. There is a more thorough discussion of this in the Ligett et al paper. Randomness plays no part in Figure 1; we instead consider the pDP losses conditioned upon the output $\hat{\theta}^P$. However, had we actually run Algorithm 1, it is unlikely (very small $\delta$) that the output $\hat{\theta}^P$ would differ so drastically (large $\omega$) from the non-private classifier $\hat{\theta}$.
>
>
> **6.** The expression $b(...)$ is the realized noise vector $b$, re-expressed as a function of $\hat{\theta}^P$ and $D$ to emphasize that it depends on sensitive data. The reason for expressing $b$ this way is mainly due to the way Theorem 6 is proved -- specifically, Equation (3) demonstrates that at the private minimizer $\hat{\theta}^P$, the noise vector is equal to the negative gradient of the non-private objective function evaluated at the same point. For GLMs (as in Corollary 7), it is easier to express this quantity directly as $\nabla J(\hat{\theta}^P; D)$, but in the general case of a convex loss function it was tricky to find a clean expression for it.
>
> **7.** Lemma 20 (in Appendix C.1) defines $F_{\lambda_1 \text{ of GOE}}$ as the CDF of the largest eigenvalue of the standard GOE (Gaussian Orthogonal Ensemble) matrix, whose distribution is calculated exactly by Chiani et al.  We will make sure we explain this in the main paper and clean up the notation.
>
>
> **8.** In Definition 3, the LHS is the *absolute value* of the log ratio so both directions are bounded (this is discussed a bit more explicitly in the proof of Theorem 6 given in Appendix D.1). Definition 2 is equivalent to the corresponding pDP definition given in Wang’s “Per-instance Privacy” paper.
>
>
> **9.** Algorithm 1 is somewhat less involved than the Obj-Pert mechanism presented in Kifer et al’s “”Private Convex Empirical Risk Minimization and High-dimensional Regression” paper. The main difference is that we only consider adding Gaussian noise, whereas Kifer et al’s algorithm also considers adding Gamma noise (for pure DP).
>
>
> **10.** “Privacy target $z$” is supposed to evoke the language of Wang’s “Per-instance Differential Privacy” paper, which uses the same expression. It’s interchangeable with “individual $z$” or “data point $z$”.
>
>
> **11.** That is a typo -- it should read “to ensure that the objective function is $\lambda$-strongly convex”.
>
>
> **12.** Yes, that’s a typo! It should be $\theta \sim \mathcal{N}(0, I_d)$.
>
>
> **13.** It’s admittedly not a great description of how we generated the synthetic data. For the logistic regression experiments, we sampled $\theta$ and $x_i$ s.t. $i \in [n]$ from the standard normal distribution $\mathcal{N}(0, I_d)$, normalizing each $x_i$ so that $||x_i||_2 = 1$ . Then each $y_i$ was determined as $y_i = 1$ if sigmoid($x_i^T \theta$) > 0.5; 0 otherwise. We’d originally toyed with adding noise to the $y_i$’s, but ultimately decided against it (only DP noise is added in the experiments on synthetic data). So ultimately, there is no $E$ noise matrix and $X \in \mathbb{R}^{n \times d}, \theta \in \mathbb{R}^d$.
>
>
> **14.** Please refer to the $(\epsilon,\delta)$-DP column of Table 1 (summary of results) from Bassily et al. 2014. The minimax rates for the excess empirical risk for the general convex + Lipschitz setting is $\tilde{O}(\frac{\sqrt{d}}{\epsilon})$ and that for the strongly convex + Lipschitz setting is $\tilde{O}(\frac{d}{n \epsilon^2 \Delta})$ where $\Delta$ is the strong convexity parameter of each individual loss function, and $n\Delta$ is the strong convexity parameter of the entire objective function.
>
> Our argument is that if we artificially add additional regularization by choosing $\lambda = n\Delta \asymp \sqrt{d}/\epsilon$  then it recovers the minimax rate for the general convex ERM problem.  Of course this changes the non-private optimal solutions by introducing a bit of bias, which can be bounded by $\lambda = \sqrt{d}/\epsilon$ too.  We will provide the above more detailed clarification in the paper.
>
>
> **15.** Good point! Throughout the paper, the unspecified norms are $\ell_2$.
>
>
> **16**. Notice that $\tilde{\epsilon}$ in the algorithm block is a tuple of two values. The first indicates the released upper bound of the ex-post pDP loss of ObjPert, the second indicates the additional pDP loss for the privacy report itself.  Therefore it should be $\bar{\epsilon}^P_1$ --- the first element of the tuple --- that appears in the middle of the theorem statement.

---

> > ### Author Response · Authors · 2021-09-06
> > **Thank you for your follow-up comments (and for raising the score)!**
> >
> > (Sorry for the delayed response. We just saw the updated part of your review.) Thank you for acknowledging that many of your technical concerns are resolved and for raising the score.
> >
> > As you pointed out, this paper does not concern the problem of calibrating noise to achieve a pre-defined worst-case DP bound (in either a data-independent or data-dependent manner).  In our opinion, fixing the randomized algorithm and privately publishing a data-dependent $\epsilon$  and fixing $\epsilon$ while changing the algorithms in "stable" ways to achieve this $\epsilon$ are two sides of the same coin.  In this paper, we address the former and show that this data-dependent $\epsilon$ can sometimes be more fine-grained  (ex-post as in "Accuracy First",  and per-instance as in Wang et al.).
> >
> > The analogy of the issue about "privacy leakage through the noise level" that one typically observes in a data-dependent DP algorithm design is replaced by the "privacy leakage through a data-dependent privacy loss", which is handled by requiring the pDP loss function to be privately published.
> >
> > We completely agree with the reviewer that using the techniques developed for propose-test-release-style algorithm design (the latter problem above) is a promising future direction.  In terms of why a joint-pDP style privacy report is necessary, we believe it enables cleverer algorithm design (e.g.,  dropping subsets of the dataset early in iterative algorithms when their per-instance DP losses hit a predefined limit). One existing work on this direction is the excellent work by Feldman and Zrnic "Individual Privacy Accounting via Renyi Filter".  Again this is another promising future direction.

---

### Official Review · Reviewer_eTH4 · 2021-07-18

**Rating:** 5
**Confidence:** 3

**Summary:**

The paper considers the problem of publishing the per-instance privacy loss of a trained learning model to each individual while protecting the privacy of other participants. The paper focuses on training a convex, smooth model using objective perturbation. By deriving bounds on the per-instance ex-post privacy loss, the paper develops algorithms that publish the Hessian matrix privately, which can be used by each individual to evaluate their per-instance privacy loss.

The paper shows that under certain regimes, the algorithm can provide an approximation of the per-instance privacy loss up to constant factors. The paper also provides empirical experiments which show how the derived method provides an approximation of the per-instance privacy cost and how it compares to the worst-case bound.

**Limitations And Societal Impact:**

Yes.

**Main Review:**

Main comments:

1. The main technical contribution of the paper is a per-instance DP analysis of the objective perturbation algorithm for empirical risk minimization. The result is OK but the analysis seems to be limited to the objective perturbation algorithm. For publishing privacy loss, the proposed algorithm (Algorithm 2) is limited to generalized linear models. How the technique can be generalized to other problems is less clear.

2. The motivation for publishing the per-instance privacy cost needs some more justification. Since this privacy loss is only observed by the user, it is cannot be used for privacy budgeting or utility improvement. It would be nice if the authors can provide some more motivation or a practical scenario where the technique can be most beneficial.

3. In the experimental results, is the privacy cost for sharing the per-instance DP loss considered in the plot? How would this affect the accuracy of the approximation?

Minor comments:
1. Line 285: I believe it should be z_n at the end.

The paper is nicely written overall and has some nice ideas. I am tending towards a rejection due to the above reasons. I am happy to increase my score if the comments can be addressed.


**Time Spent Reviewing:**

4

---

> ### Author Response · Authors · 2021-08-10
> **Author Response**
>
> Thank you for the review, and your thoughtful comments and questions. We’ve tried to address these below.
>
> **1. The main technical contribution of the paper is a per-instance DP analysis of the objective perturbation algorithm for empirical risk minimization. The result is OK but the analysis seems to be limited to the objective perturbation algorithm. For publishing privacy loss, the proposed algorithm (Algorithm 2) is limited to generalized linear models. How the technique can be generalized to other problems is less clear.**
>
> While the paper’s main result is a per-instance DP analysis of objective perturbation, we also present a pDP analysis of the Gaussian mechanism in Appendix E, as well as an improved "Analyze Gauss" mechanism in Appendix C. Theorem 22 and Corollary 23 allow us to calculate the additional pDP losses of revealing $\epsilon_1^P(\cdot)$  via the data-dependent approach. However, the pDP analysis is of general applicability to any mechanism that adds Gaussian noise. Since we have analyzed the pDP loss of the Gaussian mechanism, we can also apply the composition theorem from Wang’s “Per-instance Differential Privacy” paper to get the total pDP for a sequence of Gaussian mechanisms.
>
> The pDP analysis of the Gaussian mechanism is a fairly straightforward result compared to the pDP analysis of the objective perturbation mechanism. An interesting property of the Obj-Pert analysis is that the pDP losses depend on the output of the computation — whereas for output perturbation via the Gaussian or Laplace mechanism, conditioning on the output makes no difference. This motivates the *ex-post* setting of the paper.
>
> For Algorithm 2, the restriction to generalized models is mostly so that we can use a data-independent bound for the third term of Algorithm 1’s pDP loss $\epsilon_1(\cdot)$ which does not depend on dimension *d* (as illustrated in Theorem 9).  Since submitting the paper, we have modified the privacy report so that the data-dependent approach adds noise not just to the Hessian (to release the first term of the pDP loss $\epsilon_1(\cdot)$ of Algorithm 1), but also to the gradient (to release the third term of the pDP loss). This change allows the third term to be released without the GLM assumption; Algorithm 2 could be fully generalized to non-GLMs by using the more general bound given in Theorem 9 for releasing the first term. Note that neither the data-dependent nor -independent bounds on the first term require the GLM assumption.
>
> **2. The motivation for publishing the per-instance privacy cost needs some more justification. Since this privacy loss is only observed by the user, it is cannot be used for privacy budgeting or utility improvement. It would be nice if the authors can provide some more motivation or a practical scenario where the technique can be most beneficial.**
>
> The introduction (lines 27-40) motivates why we want to share the pDP losses. To put it another way: In most use cases, accuracy is more important than privacy. This is the same practical assumption that motivated Ligett et al’s “Accuracy First” paper, in which the data analyst wishes to maximize privacy given a fixed accuracy constraint. Per-instance DP could be valuable in a similar setting where $\epsilon$ is determined according to a desired level of utility, with the key difference being that we propose to adaptively analyze the privacy loss to particular individuals after the computation ends.
>
> Notice that it has recently become more common for researchers to use privacy parameters as high as $\epsilon = 8$ or $\epsilon=16$ in empirical DP work, and then justify that these seemingly pointless privacy guarantees are actually much stronger in practice by conducting membership inference attacks (see, e.g., https://arxiv.org/abs/2009.10031 and https://arxiv.org/abs/2106.09352).  Publishing pDP losses provides a better and theoretically sound alternative to the empirical approach for revealing the gap between the worst-case DP bound and the actual privacy loss in practice.
>
> We are also considering how pDP could be useful in designing data-dependent DP algorithms, e.g. combining the framework with a “propose-test-release” approach.
>
> **3. In the experimental results, is the privacy cost for sharing the per-instance DP loss considered in the plot? How would this affect the accuracy of the approximation?**
>
> As is, the additional privacy cost of sharing the pDP losses using the data-dependent approach isn’t considered in the experimental plots. Since submitting the paper to NeurIPS, we have modified the plots to include these additional pDP losses. As discussed in our response to comment (2),  we also modified the privacy report so that the data-dependent approach adds noise to the gradient (to release the third term of the pDP loss $\epsilon_1(\cdot)$ of Algorithm 1) as well as the Hessian (to release the first term). In our experiments with linear and logistic regression on several datasets, the data-dependent approach gives a tighter (more accurate) bound on Algorithm 1’s pDP losses, but the data-independent approach is a more practical choice after factoring in the additional pDP costs of releasing $\overline{\epsilon_1^P}(\cdot)$ via the data-dependent approach.
>
> We speculate that this might not be the case for non-GLMs, for which releasing the third term via a data-independent bound would have a necessary dependence on dimension *d* (see Theorem 9). Then we would expect to see that — especially for higher-dimensional data — the data-dependent bound is tighter even when including the additional privacy costs of adding noise to the Hessian and the gradient. Furthermore, the data-dependent bound on the first term of $\epsilon_1(\cdot)$ would be tighter when the Hessian is well-conditioned, compared to the data-independent release of the first term which uses a more conservative bound based only on the value of the regularization parameter $\lambda$.

---

> > ### Comment · Reviewer_eTH4 · 2021-09-06
> > **After rebuttal**
> >
> > Thanks a lot for the reply. It addresses some of the concerns I have. However, my concern about motivation still stands.
> >
> > The authors mentioned two motivations in the paper and the rebuttal, which includes the “Accuracy First” approach and "revealing the gap between the worst-case DP bound and the actual privacy loss in practice". However, it doesn't seem the proposed approach serves either of the purposes. As pointed out in the paper, the published pDP parameter is only accessible to the user themself since the evaluation needs users' original clean data (This is also hinted by the joint DP guarantee). Hence the published pDP parameter cannot be used to do privacy evaluation as in the “Accuracy First” approach or analyze the gap between the worst-case DP bound and the actual privacy loss as the service provider (learner) has no access to it.
> >
> > Given this, I would keep my score on the paper. I would suggest the authors provide more discussion on the motivation as well as limitations on the proposed approach in future revisions.

---

> > > ### Author Response · Authors · 2021-09-06
> > > **The published pDP parameter is a function of any individual (rather than only those in the dataset)**
> > >
> > > Thank you for acknowledging our response!
> > >
> > > And thank you for further clarifying the question on motivation. We believe what you have in mind on "the actual privacy loss" is one that is *independent to the individuals* themselves, but could *depend on the dataset as a whole* if privately published  (as in the data-dependent DP analysis in Papernot et al. 2018).
> > >
> > > We wanted to kindly point out that our approach also implies the (private) release of such an "actual privacy loss" by your definition.  Recall that we propose to privately publish a high probability upper bound of the ex post pDP function --- $\tilde{\epsilon}(z)$. This is a function that can take any individual $z$ in the population $\mathcal{Z}$, whether or not this individual is in the dataset or not.  When we plug in individuals in the dataset, the tuple  $(\tilde{\epsilon}(z_1),...,\tilde{\epsilon}(z_n))$ satisfies joint-(per-instance)-DP.  But we could also take supremum over all $z\in\mathcal{Z}$,  which we recover what the reviewer has in mind by $\sup_z \tilde{\epsilon}$,
> > >
> > > In this sense, our approach is strictly more general. We hope the above clarifies your concern.

---

### Official Review · Reviewer_aTij · 2021-07-23

**Rating:** 7
**Confidence:** 3

**Summary:**

The paper considers the problem of bridging the gap between central DP, where overall epsilon is released, versus personalized epsilon, that is privacy of individual data points that may or may not have been part of a dataset analyzed with central DP.
The paper considers central DP of specific function: objective perturbation mechanism, this setting of achieving central DP is less common that others.
The paper makes an observation that naively computing personalized epsilon and releasing it will violate privacy of central DP (as this computation depends on private data) as anyone can use such an oracle to determine the real value that central DP is trying to hide.
Since the paper considers a specific DP mechanism its results are applied only to private empirical risk minimization problems (ie function computed on the private dataset).

The proposed mechanism works by computing personalizes privacy loss (es-post pDP) directly (since it considers convex problems it is possible). It then adds noise to computed result because otherwise this metric cannot be released. However the paper also derives data-independent upper bounds for pDP that do not incur additional noise.


**Limitations And Societal Impact:**

Limitations are pointed out in one sentence in conclusion.

**Main Review:**

This is a very nice paper and I really enjoyed reading it.

My main concern is with it applicability to other DP settings (e.g., Laplace mechanism) and other mechanisms that rely on Gaussian noise and do not build on optimization function perturbation.

Experimental section could  additional experiments, including experiments showing distinction between the data independent bound and the bound that contains additional Gaussian noise added, and real datasets.



**Time Spent Reviewing:**

3

---

> ### Author Response · Authors · 2021-08-10
> **Author Response**
>
> Thanks for your review! You’ve made some good points which we’ll address below.
>
> **>>> My main concern is with it applicability to other DP settings (e.g., Laplace mechanism) and other mechanisms that rely on Gaussian noise and do not build on optimization function perturbation.**
>
> The paper’s main results pertain to objective perturbation, but the appendix includes a section which analyzes the pDP loss of the Gaussian mechanism in Appendix E. (In Appendix C, we also present an improved “Analyze Gauss” mechanism.) We use Corollary 23 primarily to calculate the pDP losses of privately releasing the Hessian $\hat{H}^P$ (Algorithm 6)  and the gradient $g^P$ (Algorithm 5), which demonstrate how the per-instance privacy analysis can be applied to mechanisms that use Gaussian noise. Since we have analyzed the pDP loss of the Gaussian mechanism, we can also apply the composition theorem from Wang’s “Per-instance Differential Privacy” paper to get the total pDP for a sequence of Gaussian mechanisms.
>
> The pDP analysis of the Gaussian mechanism is a fairly straightforward result compared to the pDP analysis of the objective perturbation mechanism. An interesting property of the Obj-Pert analysis is that the pDP losses depend on the output of the computation — whereas for output perturbation via the Gaussian or Laplace mechanism, conditioning on the output makes no difference. This motivates the *ex-post* setting of the paper.
>
> **>>> Experimental section could additional experiments, including experiments showing distinction between the data independent bound and the bound that contains additional Gaussian noise added, and real datasets.**
>
> Since submitting the paper to NeurIPS, we have run additional experiments including those that compare the data-independent and data-dependent approaches. We also modified the privacy report so that the data-dependent approach adds noise not just to the Hessian (to release the first term of the pDP loss $\epsilon_1(\cdot)$ of Algorithm 1), but also to the gradient (to release the third term of the pDP loss).
>
> In our updated experiments with linear and logistic regression on several datasets, the data-dependent approach gives a tighter (more accurate) bound on Algorithm 1’s pDP losses, but the data-independent approach is a more practical choice after factoring in the additional pDP costs of releasing the data-dependent bound via Algorithm 2.
>
> We speculate that this might not be the case for non-GLMs, for which releasing the third term of $\epsilon_1(\cdot)$ via a data-independent bound would have a necessary dependence on dimension *d* (see Theorem 9). Then we would expect to see that — especially for higher-dimensional data — the data-dependent bound is tighter even when including the additional privacy costs of adding noise to the Hessian and the gradient. Furthermore, the data-dependent bound on the first term would be tighter when the Hessian is well-conditioned, compared to the data-independent release of the first term which uses a more conservative bound based only on the value of the regularization parameter $\lambda$.

---

### Decision · Program_Chairs · 2021-09-27

**Decision:**

Accept (Poster)

**Comment:**

The authors thought that data dependent privacy properties are an interesting direction of study. The main concern in discussion was the motivation for the results. The reviewers questioned when this notion of privacy would be useful, commenting (among other things) that it could not be directly used by the algorithm designer, only reported to the users. There were also comments about the readability, which could be improved. The paper otherwise seems solid.

As this is a relatively new and underexplored notion, I am OK with accepting the paper even if it doesn't fulfil the full promise discussed at the start of the intro. However, by the same token that this is a new notion, it is very important that the authors address this motivation better than done in the current submission: for instance, identifying and commenting on the strengths and weaknesses of the result (or any results of this form). The should also be better discussion of related work, how this notion differs from very similar ones, relative advantages, etc.: it may be tough for an outsider to distinguish between the several similar sounding notions. I expect that the authors will do all this in their final version of this paper.